# Towards omics-based predictions of planktonic functional composition from environmental data

Emile Faure [1,2 ✉], Sakina-Dorothée Ayata[1,2,4] & Lucie Bittner [2,3,4]

Marine microbes play a crucial role in climate regulation, biogeochemical cycles, and trophic networks. Unprecedented amounts of data on planktonic communities were recently collected, sparking a need for innovative data-driven methodologies to quantify and predict their ecosystemic functions. We reanalyze 885 marine metagenome-assembled genomes through a network-based approach and detect 233,756 protein functional clusters, from which 15% are functionally unannotated. We investigate all clusters' distributions across the global ocean through machine learning, identifying biogeographical provinces as the best predictors of protein functional clusters' abundance. The abundances of 14,585 clusters are predictable from the environmental context, including 1347 functionally unannotated clusters. We analyze the biogeography of these 14,585 clusters, identifying the Mediterranean Sea as an outlier in terms of protein functional clusters composition. Applicable to any set of sequences, our approach constitutes a step towards quantitative predictions of functional composition from the environmental context.

[1] Sorbonne Université, CNRS, Laboratoire d'Océanographie de Villefranche, LOV, Villefranche-sur-Mer, France. [2] Institut de Systématique, Evolution, Biodiversité (ISYEB), Muséum National d'Histoire Naturelle, CNRS, Sorbonne Université, EPHE, Université des Antilles, Paris, France. [3] Institut Universitaire de France, Paris, France. [4]These authors contributed equally: Ayata Sakina-Dorothée, Bittner Lucie. ✉email: emile.faure@univ-brest.fr

Planktonic organisms play an essential role in biogeochemical cycles through the capture and export of carbon into the deep ocean, nitrogen fixation, remineralization of organic matter, or the production of dimethyl-sulfur, hence impacting global climate[1–5]. The understanding and modeling of such geochemical functions is key for predicting the global functioning of oceanic ecosystems, and especially their response to climate change[6–8]. These biogeochemical functions are usually modeled by simulating the dynamics of plankton functional types (PFT) that are theoretical entities grouping planktonic organisms according to shared functional capacities (e.g., calcifiers, nitrogen fixers, or silicifiers)[6]. This approach allows to incorporate the functional diversity of marine plankton into biogeochemical models[8–11] but often relies on a priori and restricted choices of the considered types of planktonic organisms and of their physiological rates or parameters[12]. For example, bacteria are often lacking an explicit representation in global PFT models[9,11], even though more than $10^{30}$ bacterial cells inhabit the ocean's subsurface[3]. To tackle this issue, recent works proposed to switch towards data-driven modeling of planktonic communities and their impact on the environment, notably through the use of high-throughput sequencing data[10,13–16].

Next-generation sequencing technologies have led to significant advances in the knowledge of the taxonomic and functional diversity of planktonic organisms[5,17,18]. Bioinformatics workflows allow the assembly of metagenome-assembled genomes (MAGs), which are near-complete genomes retrieved from DNA fragments coming from environmentally sequenced individuals of one or a few closely related populations[19–22]. MAGs can be taxonomically annotated using multi-marker gene approaches, and organism-level functional profiles can be drawn from their genomic content[19–21]. Reads from environmental meta-omics datasets can also be mapped to their reconstructed sequences to obtain abundance measurements both at MAG and single protein level[21–23]. MAGs can be considered as representative of the genetic potential of natural populations, hence allowing to retrieval of genomes of cultivable, uncultivable, or even unknown species present in the environment. They constitute a promising tool for investigating as a whole the functional potential of known and unknown planktonic life forms.

Recently, a genomics-based model revealed that the gene content of planktonic communities is more relatable to biogeochemical gradients than taxonomic content[10]. In another study, omics data were used to quantitatively estimate global nitrogen fixers abundance through machine learning algorithms[24]. It illustrates how quantitative, data-driven biogeochemical models can be built from global omics datasets. However, these studies focused only on a relatively small number of well-described genes (e.g., *nif* or *amtB* genes, involved in dinitrogen and ammonium fixation, respectively)[10,24], far from exploiting the rich functional diversity observed in omic samples. In this way, the large proportion of unknown sequences detected in environmental meta-omics datasets, that is to say, the open reading frames (ORFs) which can not be linked to any biological functions (usually around 40% for bacteria and archaea, and about 50% for eukaryotes), is as yet untapped[4,5,23,25–27]. Besides, many meta-omics studies have either focused on semi-quantitative diversity and interactions surveys at global scales[25,28], on specific taxonomic groups (e.g., Collodaria)[29], or on particular biological functions (such as nitrogen fixation or mixotrophy)[21,24,30]. A recent study has grouped protein sequences of marine planktonic bacteria and archaea according to their annotated metabolic pathways to investigate their differential abundance and expression, mainly focusing on pre-selected biogeochemical functions such as photosynthesis or nitrogen fixation[23]. By investigating the response of biogeochemistry-related protein groups to environmental conditions, significant differences in terms of presence and expression were identified between polar and non-polar areas, and between mesopelagic and surface depths[23]. These results highlight the potential of function-clustering-based approaches for deciphering global ocean biogeochemistry but could be further extended by skipping any sequence pre-selection step requiring database-dependent metabolic pathways annotations.

In this study, we followed a similar approach while avoiding any a priori choices of particular genes or metabolic pathways. We used 51 quantitative and qualitative environmental variables to detect both known and unknown protein clusters that are sensitive to environmental gradients. We re-analyzed 885 high-quality MAGs from marine planktonic Bacteria ($n = 820$) and Archaea ($n = 65$), assembled by Delmont et al.[21]. using 93 *Tara Oceans* picoplanktonic metagenomes from the surface of the global ocean. With these almost 2 million sequences, we built functional clusters of proteins using a sequence similarity network (SSN), i.e., a graph in which nodes are protein sequences, and edges represent the similarity and coverage between each pair of sequences[31–35]. Such approaches allow for the construction of sequence clusters putatively homogenous in function[31] and were recently used to investigate the genomic basis of functional diversity in bacteria and archaea[36], in a lineage of eukaryotes[33], or in natural microbial communities[35]. Particularly, we are here interested in knowing if the abundance of some protein clusters could be predicted from environmental data in the oceanic ecosystem. For example, is the distribution of biogeochemistry-related protein clusters more sensitive to environmental gradients than one of the other clusters? We thus explored the biogeography of environment-related protein clusters in light of their potential functional and/or taxonomic annotation, in order to identify the ones being specific to certain environmental conditions, such as oligotrophic or particularly cold waters.

We introduce here a data-driven, large-scale, fast, and automatable approach, potentially applicable to any set of environmental sequences, which involves (1) the network-based construction of sequence clusters, putatively homogeneous in function, (2) the functional annotation of these clusters, (3) the calculation of environmental abundance values for each of these protein clusters through environmental reads re-mapping, and (4) the description of statistical relationships between cluster abundances and environmental gradients through machine learning and constrained ordination methods. We then present a biogeographical analysis of known and unknown bacterial and archaeal protein functional clusters (PFCs) identified as sensitive to environmental gradients in the global ocean, with no a priori choice of specific functions or taxa. Particularly, we investigate the biogeography of 14,585 PFCs from which the abundance is predictable from the environmental context. We identify biogeographical provinces as the best predictors of PFCs distribution, and the Mediterranean Sea as an outlier in terms of PFCs composition. Our results demonstrate the potential of omics-based predictions of planktonic communities functional composition based on environmental data.

## Results

**From SSN to PFCs.** We analyzed the 1,914,171 proteins from 885 MAGs from marine plankton, recovered from 12 geographically bound assemblies of metagenomic sets corresponding to a total of 93 Tara Oceans samples from the 0.2 to 3 μm and 0.2 to 1.6 μm size fractions[21]. A flowchart of our bioinformatic pipeline is available in Supplementary Fig. 1. 39.6% of the MAGs' proteins (757,457) were involved in our SSN, i.e., they had at least one similarity relationship with another protein that satisfied the chosen threshold of 80% similarity and 80% coverage (see "Methods").

In total, 51.1% of the network proteins could be annotated to 4922 unique molecular function IDs in the KEGG database[37], associated with 327 distinct metabolic pathways (a full list of these pathways is available in Supplementary Data 1). In total, 85.2% of the network proteins were annotated to 17,009 eggNOG functional descriptions[38,39].

The SSN involved 233,756 connected components (CCs), i.e., groups of nodes (here proteins) connected together by at least one path and disconnected from the rest of the network. According to KEGG and eggNOG databases, 15.3% and 48.5% of the CCs remained without any functional annotation (i.e., all sequences from the CC were unmatched in the databases, or had a match but were not yet linked to any biological function, Table 1), and 14.8% were functionally unannotated for both databases. We ranked the functional homogeneity of CCs involving at least one functional annotation from 0 (all annotations in the CC are different) to 1 (all annotations in the CC are the same) and found mean homogeneity scores of 0.99 over 1 for KEGG annotations and 0.94 over 1 for eggNOG ones (see "Methods" for score calculation details). Only 88 (0.04%) CCs had a homogeneity score below 0.5 in both annotation databases, all with sizes below five proteins. 177 CCs (0.07%) had a score below 0.8 in both databases, all under 12 proteins in size. These CCs were kept in the analysis while tagged as poorly homogenous. We thereafter considered each CC as a PFC, numbered from #1 to #233,756.

To check for the influence of taxonomic relationships between the MAGs on our PFCs, we computed different metrics based on MAGs taxonomic annotations provided by Delmont et al.[21]. (Table 1). This taxonomic annotation based on 43 single-copy core genes allowed to annotate 100% of the MAGs at the domain level, and 95% of the MAGs at the phylum level, the remaining 5% corresponding to Bacteria MAGs of unidentified phyla[21]. Only 1330 PFCs (0.6%) mixed proteins from the Archaea and Bacteria domains. PFCs were very homogeneous at the phylum level, then the homogeneity decreased at lower taxonomic rank, meaning that PFCs studied here were generally not specific from a single class, order, family, genus, or MAG (Table 1). In total, 7834 PFCs (3.4%) were only composed of proteins with no functional annotation in KEGG and eggNOG databases, and no taxonomic annotation under the phylum level. Their sizes ranged from 2 to 30 proteins (mean of 2.62). Their 20,552 proteins came from Euryarchaeota MAGs (12,458; 60.6%), Bacteria MAGs of unidentified phylum (2742; 13.3%), Candidatus Marinimicrobia MAGs (2451; 11.9%), Proteobacteria MAGs (1528; 7.4%), Acidobacteria MAGs (1031; 5%), Verrucomicrobia MAGs (103; 0.5%), Planctomycetes MAGs (89; 0.4%), Bacteroidetes MAGs (79; 0.4%), Chloroflexi MAGs (59; 0.3%) and Candidate Phyla Radiation MAGs (12; 0.05%). We hereafter considered these functionally and taxonomically unknown PFCs as "dark" PFCs[40,41]. Their nucleotidic sequences are available in separate supplementary files (see "Data availability"). The abundance of dark PFCs was significantly different from the abundance of other PFCs in 85 samples over 93 (two-sided Wilcoxon rank-sum test, $p$-value < 0.05). The median abundance of dark PFCs was higher than the one of other PFCs in 36 of these 85 samples, and lower in the 49 others. Further details on dark PFCs' abundances are available in section I of Supplementary notes.

**Identification of PFCs highly related to environmental gradients**. To identify the PFCs that responded the most to environmental gradients, we first selected the 228,914 clusters with non-zero variance abundance profiles (i.e., at least 10% of distinct abundance values across all samples, and less than a 95 to 5 ratio between the most and the second most observed abundance value, please see "Methods" for more details), to avoid the creation of constant or near-constant training and/or test sets. We then built random forest regression models for each of these 228,914 clusters. We used the sequence abundances as response variables or labels, and 52 environmental variables as explanatory variables (see "Methods" for details of model training and tuning). More than half of the random forest regression models showed a clear statistical signal: 130,651 models (55.9%) had $R^2$ values over 0.25, corresponding to PFCs linked to environmental conditions, and 14,585 (6.4%) had $R^2$ values over 0.5 (Fig. 1A), corresponding to PFCs highly linked to environmental gradients (hlePFCs), from which the abundances were potentially predictable from the environmental context (Fig. 1B). The mean $R^2$ value over all models was 0.29, with a maximum of 0.88 (Fig. 1A). Longhurst biogeographical provinces[42] were detected as the most important predictor in 98,450 models (43.0%) and were in the top three most important predictors in 167,039 models (73.0%) (Fig. 1C). Among models with biogeographical provinces as the best predictor, the mean $R^2$ reached 0.30. The temperature was in the top three most important predictors in 12,673 models (5.5%). Models with temperature as the best predictor had a mean $R^2$ of 0.43, which was the fourth-highest value of all quantitative variables, behind sunshine duration (0.49), ammonium at 5 m depth (0.46), and annual density (0.43).

We focused on the 14,585 PFCs associated with models showing $R^2$ values over 0.5, hereafter called "hlePFCs" for hlePFCs PFCs. 246 KEGG pathways were associated with the 14,585 hlePFCs, i.e., 75% of the pathways identified on the full network were detected in hlePFCs. Supplementary Data 1 gives a detailed list of all pathways detected in our network, along with their total number of occurrences in PFCs, and their number of occurrences in hlePFCs.

The functional homogeneity of the 14,585 hlePFCs was similar to the one of the total 233,756 PFCs (Tables 1, 2). In parallel, 9.2% of the hlePFCs were functionally unannotated (e.g., only composed of unannotated proteins in the KEGG and the eggNOG functional databases), while 14.8% of the PFCs were functionally unannotated.

Proportions of taxonomically homogeneous PFCs were similar between hlePFCs (Table 2) and total PFCs (Table 1), all above 90% at the phylum, class, order, and family level when considering only PFCs including at least one protein taxonomically annotated. At the genus level, the proportion of taxonomically homogeneous PFCs decreased from 91.9% in total PFCs to 66.6% in hlePFCs (Tables 1 and 2). Hence, hlePFCs tended to be shared by more genera compared to all PFCs, while retaining a high functional homogeneity. The proportion of taxonomically unannotated hlePFCs was lower than the one of total PFCs at the phylum and class levels but was higher at the order, family, and genus levels (Tables 1 and 2).

The mean $R^2$ values of models associated with the 7834 dark PFCs (i.e., PFCs without any functional annotation and without taxonomic annotation below the phylum level) was of 0.27, and 166 (i.e., 2.1%) of them were selected among the 14,585 hlePFCs. These 166 dark hlePFCs corresponded to 357 proteins belonging to 51 unique MAGs, annotated as Proteobacteria (8 MAGs, 186 proteins), Euryarchaeota (19 MAGs, 40 proteins), Candidatus Marinimicrobia (13 MAGs, 97 proteins), Bacteroidetes (2 MAGs, 2 proteins), Chloroflexi (1 MAG, 5 proteins), Verrucomicrobia (1 MAG, 2 proteins) and Bacteria unannotated at Phylum level (7 MAGs, 25 proteins). Forty-six of these 51 MAGs were estimated to be over 70% complete, and the remaining five showed completion estimates between 50 and 70%[21].

**Global biogeography of the PFCs hlePFCs**. The canonical correspondence analysis (CCA) achieved on the 14,585 hlePFCs to

**Table 1 Metrics computed on the 233,756 protein functional clusters (PFC) from the sequence similarity network of MAGs proteins.**

| PFC size | | Functional scores | | | Taxonomy scores | | |
|---|---|---|---|---|---|---|---|
| | | Homogeneity | Unknowns quantification | | Homogeneity | Unknowns quantification | |
| Mean | 3.24 | Mean homogeneity score with EggNOG annotations (Number of NA values): 0.94 (35,818) | EggNOG annotations — PFCs only composed of annotated proteins (% of total PFCs): 181,595 (77.7%) | | PFCs associated to only 1 Phylum (% of total PFCs) (% of PFCs with at least one Phylum annotation): 221,541 (94.8%) (97.5%) | Phylum level — Only proteins from annotated MAGs (% of total PFCs): 220,839 (94.5%); Only proteins from unannotated MAGs (% of total PFCs): 6,367 (2.7%) | |
| | | | PFCs with at least one annotated protein (% of total PFCs): 197,938 (84.7%) | | PFCs associated to only 1 Class (% of total PFCs) (% of PFCs with at least one Class annotation): 192,095 (82.2%) (96.8%) | Class level — Only proteins from annotated MAGs (% of total PFCs): 186,331 (79.7%); Only proteins from unannotated MAGs (% of total PFCs): 35,338 (15.1%) | |
| | | | PFCs only composed of unknown proteins (% of total PFCs): 35,818 (15.3%) | | PFCs associated to only 1 Order (% of total PFCs) (% of PFCs with at least one Order annotation): 144,265 (61.7%) (93.8%) | Order level — Only proteins from annotated MAGs (% of total PFCs): 135,046 (57.8%); Only proteins from unannotated MAGs (% of total PFCs): 79,921 (34.2%) | |
| Minimum | 2 | Mean homogeneity score with KEGG annotations (Number of NA values): 0.99 (113,321) | KEGG annotations — PFCs only composed of annotated proteins (% of total PFCs): 91,103 (39.0%) | | PFCs associated with only 1 Family (% of total PFCs) (% of PFCs with at least one Family annotation): 100,801 (43.12%) (95.3%) | Family level — Only proteins from annotated MAGs (% of total PFCs): 88,404 (37.8%) | |
| | | | PFCs with at least one annotated protein (% of total PFCs): 120,435 (51.5%) | | PFCs associated to only 1 Genus (% of total PFCs) (% of PFCs with at least one Genus annotation): 21,921 (9.4%) (91.9%) | Genus level — Only proteins from annotated MAGs (% of total PFCs): 128,010 (54.76%); Only proteins from unannotated MAGs (% of total PFCs): 13,544 (5.8%) | |
| Maximum | 1072 | | PFCs only composed of unknown proteins (% of total PFCs): 113,321 (48.5%) | | PFCs associated with only 1 MAG (% of total PFCs): 7146 (3.1%) | Only proteins from unannotated MAGs (% of total PFCs): 209,892 (89.8%) | |

Functional scores are based on the functional annotation of MAGs proteins, with a functional homogeneity score of 1 meaning that all proteins in a PFC share the same annotation, while a score of 0 indicates that all proteins have different annotations (see "Methods" for details). By "unknown proteins" we refer both to sequences with no match in databases (KEGG and/or eggNOG) and to sequences existing in databases but with no functional and/or taxonomic annotation. Taxonomy scores are based on taxonomic annotations of MAGs available from Delmont et al.[21]. This way, the 6,367 PFCs with only proteins from MAGs unannotated at the phylum level were only composed of proteins coming from the 45 Bacteria MAGs of the unidentified phylum. Detailed functional and taxonomic annotations for each protein sequence are available online, as well as detailed sizes and functional/taxonomy scores for each PFC (see "Data availability").

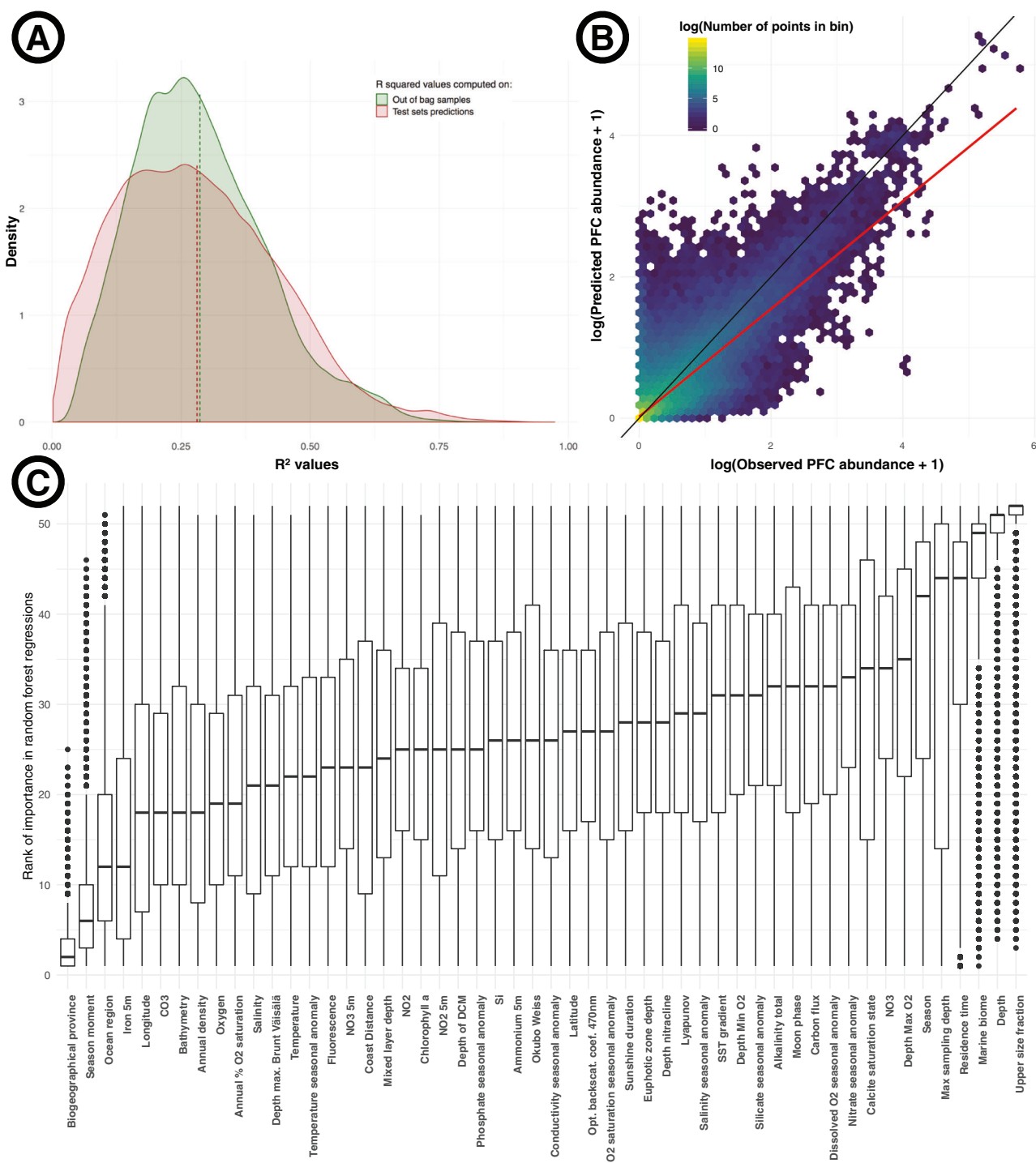

**Fig. 1 Predicting protein functional clusters abundances from environmental data. A** Density distribution of $R^2$ values of all the 228,914 random forest models. The green curve corresponds to $R^2$ values obtained using out-of-bag samples of the fivefold cross-validation process applied during model training, while the red curve corresponds to $R^2$ values obtained from predictions made on test sets using the selected trained models (See "Methods" for details on the statistical approach). The mean $R^2$ value over all models was of 0.29 using out-of-bag samples, and 0.28 based on predictions over test sets. **B** Relationship between abundances observations from test sets and the corresponding model predictions, for all the 14,585 models showing $R^2$ values above 0.5, i.e., a total of 3,174,580 predictions/observations couples. Both observations and predictions were log-transformed for graphical purposes, as the range of abundance values varied across three orders of magnitudes across different PFCs. The space of the plot was divided into hexagonal bins colored according to the number of points located in each of them, in log scale. The black line represents exact predictions, while the red line corresponds to the linear model fit between the two plotted variables ($R^2$ of 0.70). **C** The rank of importance of each environmental variable in models. Ranks were attributed from 1 for the most important to 52 for the least important variable in each model (a lower rank then indicates higher importance in models). Each boxplot summarizes the ranks of the importance of its focal variable in all the 228,914 models. Boxplots minima and maxima correspond to −1.5 and 1.5 times the interquartile range, while the limits of boxes correspond to the first and third quartiles. The centerline indicates the median. All points outside of the minima-maxima range are plotted.

**Table 2 Metrics computed on the 14,585 protein functional clusters detected as particularly linked to environmental gradients (hlePFCs).**

| PFC size | Functional scores | | Taxonomy scores | |
|---|---|---|---|---|
| | Homogeneity | Unknowns quantification | Homogeneity | Unknowns quantification |
| **Mean**: 3.6% | Mean homogeneity score with EggNOG annotations (Number of NA values): 0.94 (1403) | PFCs only composed of annotated proteins (% of total PFCs): 12,172 (83.5%); PFCs with at least one annotated protein (% of total PFCs): 13,182 (90.4%) | PFCs associated to only 1 Phylum (% of total PFCs) (% of PFCs with at least one Phylum annotation): 13,987 (95.9%) (96.8%); PFCs associated to only 1 Class (% of total PFCs) (% of PFCs with at least one Class annotation): 12,188 (83.6%) (95.7%) | Phylum level — Only proteins from annotated MAGs (% of total PFCs): 14,049 (96.3%); Only proteins from unannotated MAGs (% of total PFCs): 137 (0.1%). Class level — Only proteins from annotated MAGs (% of total PFCs): 11,814 (81.0%); Only proteins from unannotated MAGs (% of total PFCs): 1843 (12.6%) |
| **Minimum**: 2 | | PFCs only composed of unknown proteins (% of total PFCs): 1403 (9.6%); PFCs only composed of annotated proteins (% of total PFCs): 6485 (44.5%) | PFCs associated with only 1 Order (% of total PFCs) (% of PFCs with at least one Order annotation): 7600 (52.1%) (94.4%); PFCs associated with only 1 Family (% of total PFCs) (% of PFCs with at least one Family annotation): 4459 (30.6%) (94.7%) | Order level — Only proteins from annotated MAGs (% of total PFCs): 6206 (42.6%); Only proteins from unannotated MAGs (% of total PFCs): 6536 (44.8%). Family level — Only proteins from annotated MAGs (% of total PFCs): 2998 (20.6%); Only proteins from unannotated MAGs (% of total PFCs): 9876 (67.7%) |
| **Maximum**: 222 | Mean homogeneity score with KEGG annotations (Number of NA values): 0.995 (5964) | PFCs with at least one annotated protein (% of total PFCs): 8621 (59.1%); PFCs only composed of unknown proteins (% of total PFCs): 5964 (40.9%) | PFCs associated to only 1 Genus (% of total PFCs) (% of PFCs with at least one Genus annotation): 548 (3.8%) (66.6%); PFCs associated with only 1 MAG (% of total PFCs): 515 (3.5%) | Genus level — Only proteins from annotated MAGs (% of total PFCs): 367 (2.5%); Only proteins from unannotated MAGs (% of total PFCs): 13,762 (94.4%) |

Functional scores are based on the functional annotation of MAGs proteins, with a functional homogeneity score of 1 meaning that all proteins in a PFC share the same annotation, while a score of 0 indicates that all proteins have different annotations (see "Methods"). By "unknown proteins" we refer both to sequences with no match in databases (KEGG and/or eggNOG) and to sequences existing in databases but with no functional and/or taxonomic annotation. Taxonomy scores are based on taxonomic annotations of MAGs from Delmont et al.[21].

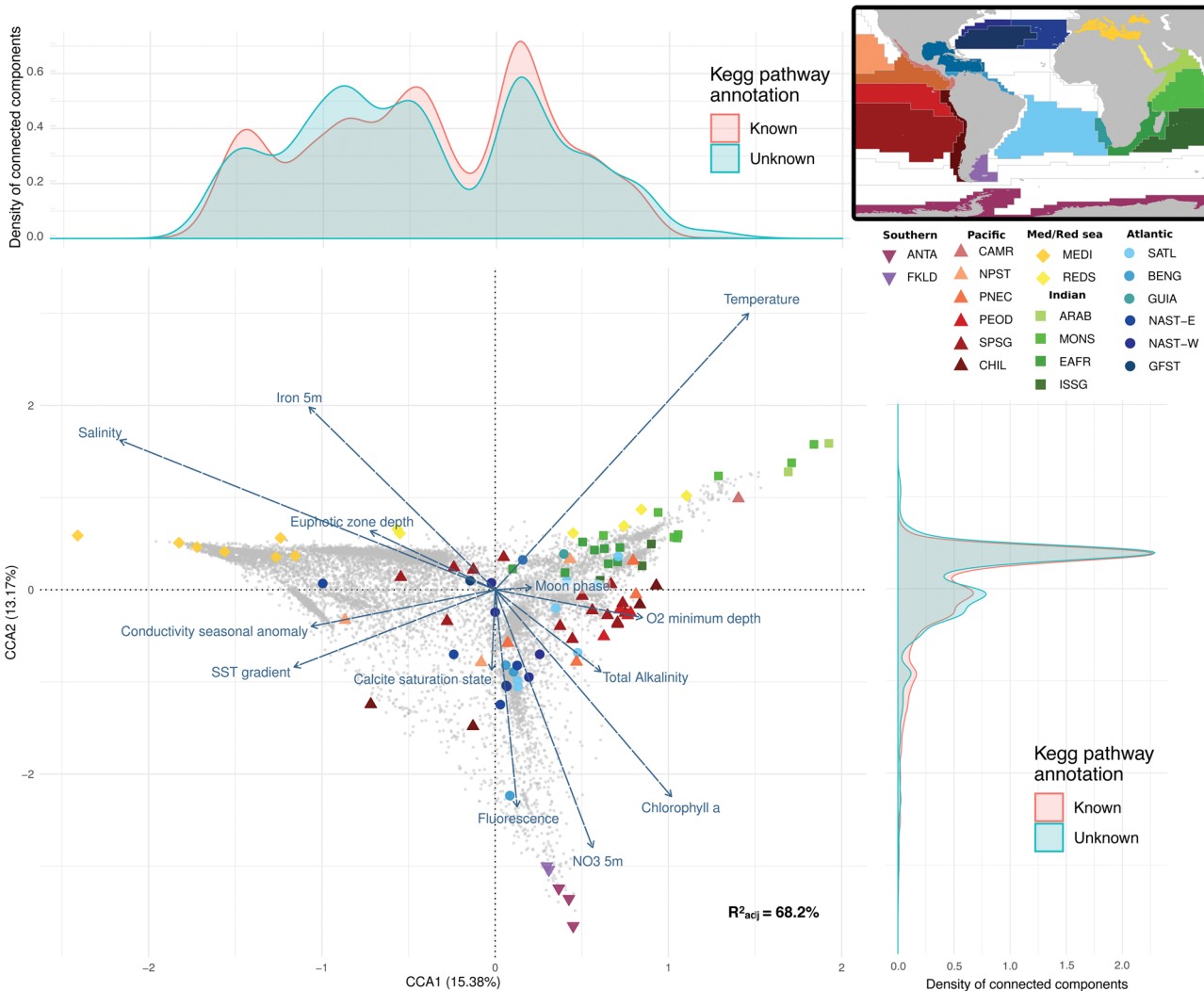

**Fig. 2 Canonical correspondence analysis (CCA) on abundances of the 14,585 protein functional clusters highly linked to environmental variables (hlePFCs).** hlePFCs are represented as gray dots, quantitative environmental variables as arrows, and samples as points colored and shaped according to their biogeographical province (correspondence between four letters codes used here and full biogeographical provinces names, as well as descriptions of all other environmental variables are available in Supplementary Data 2). A map of Longhurst biogeographical provinces[42] colored using the same color scale is shown in the upper right panel. For simplification issues, other qualitative variables (season moment, depth, and ocean region) were not represented. The closest sample from the Mediterranean ones in the CCA space, which comes from the closest Atlantic station to the strait of Gibraltar, was highlighted by a plain black arrow. On the right and upper left panels, density plots are represented along each axis, illustrating the density of functionally annotated and unannotated hlePFCs based on KEGG annotations (functionally annotated hlePFCs contain at least one functionally annotated protein; functionally unannotated hlePFCs contain only functionally unannotated proteins). The mean hlePFC density was of 0.25 along CCA1 (standard deviation = 0.2, maximum = 0.71), and 0.2 along CCA2 (standard deviation = 0.42, maximum = 2.46). The mean difference in density between functionally annotated and unannotated hlePFCs along CCA1 was of 0.02 (standard deviation = 0.07, maximum = 0.23). The mean density difference between annotated and unannotated hlePFCs along CCA2 was 0.01 (standard deviation = 0.24, maximum = 1.33). Similar observations were done using eggNOG annotations densities (Supplementary Fig. 2).

investigate their biogeographical repartition had an $R^2$ value of 68.2% and was significant ($p$-value < 0.001). The first axis (15.38% of explained variance) opposed warm samples from the Indian Ocean (CCA1 > 0) to saline samples from the Mediterranean (CCA1 < 0) (Fig. 2). The second axis (13.17%) opposed cold and nutrient-rich samples from the Southern Ocean (CCA2 > 0) to warmer and more oligotrophic samples. Samples from geographically close biogeographical provinces appeared close to each other in the CCA space, with samples from the Southern Ocean and the Atlantic zones on the bottom, the Pacific Ocean in the middle, and the Indian Ocean on the right of the CCA1-CCA2 space (Fig. 2). The closest sample from the Mediterranean ones in the CCA space was from the surface of the closest Atlantic

station to the strait of Gibraltar (station TARA_004) at the entrance of the Mediterranean Sea (sample highlighted by a black arrow in Fig. 2).

Combining the CCA results with the functional annotation of hlePFCs, we observed that the vast majority of metabolic pathways were not enriched in particular environmental conditions (Fig. 3). Among the few exceptions, *Atrazine degradation* was lightly enriched in the Mediterranean Sea (CCA10), while *RNA polymerase* and *AMPK signaling* pathways were lightly enriched in nutrient-rich, cold waters (Fig. 3). Pathways related to biogeochemical functions (e.g., *carbon fixation pathways in bacteria/archaea, Nitrogen metabolism*, or *methane metabolism*) or linked to ecological interactions between organisms (e.g., *biosynthesis of antibiotics,*

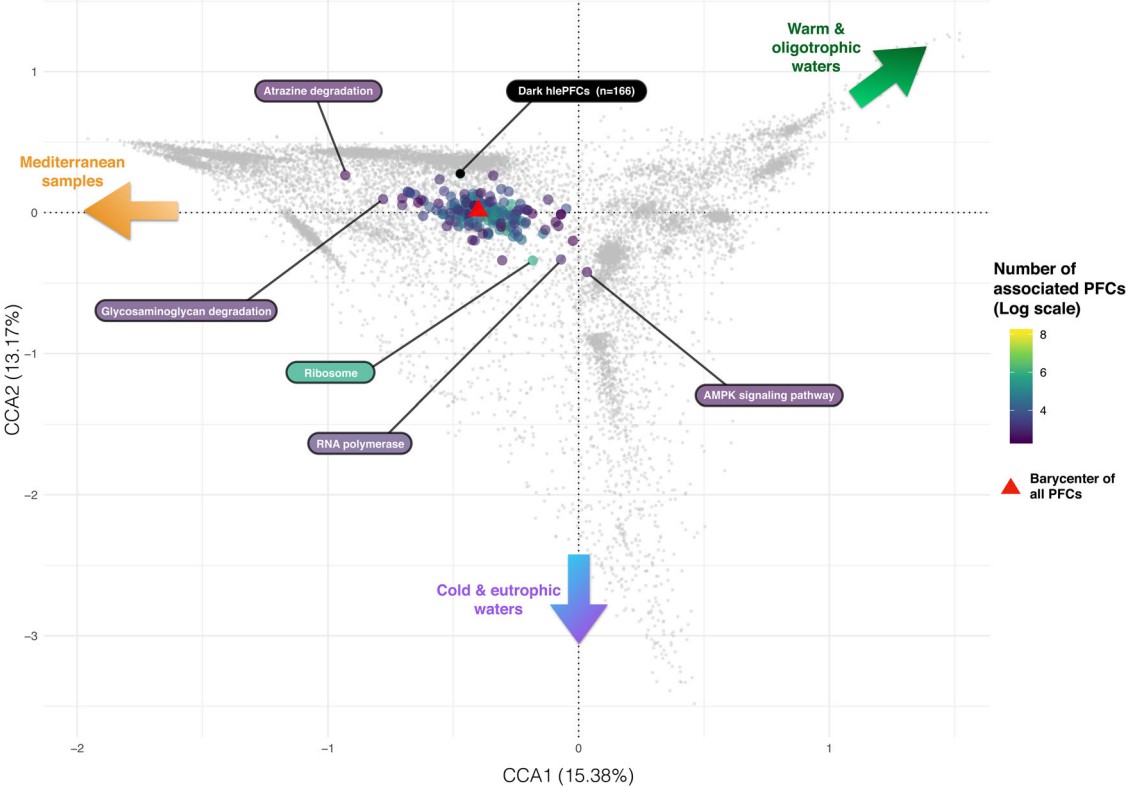

**Fig. 3 Barycenters of metabolic pathways detected at least 10 times in the canonical correspondence analysis (CCA) space.** Gray dots represent hlePFCs in the same way as in Fig. 2, while colored points indicate the barycenters of KEGG metabolic pathways that occurred at least ten times among hlePFCs, the barycenter of a pathway corresponding to the barycenter of the positions of its associated hlePFCs. The color of each barycenter codes for the number of PFCs annotated to the focal pathway, in log scale. A red triangle indicates the barycenter of all hlePFCs in the CCA space. The barycenter of dark hlePFCs (i.e., hlePFCs without functional annotation and the taxonomical assignment below the phylum level) was represented in black Colored arrows indicate the environmental conditions associated with the different zones of the CCA space (see Fig. 2).

*quorum sensing*, or *ABC transporters*) were present homogeneously in the CCA space (Supplementary Fig. 3). Functionally unannotated hlePFCs were more abundant around −1 and above 1 along CCA1 (Fig. 2), corresponding to hlePFCs associated with Mediterranean samples and Indian Ocean samples, respectively. The 166 dark hlePFCs were associated with Mediterranean samples, and almost absent from polar samples (Fig. 5B, Supplementary Fig. 3).

We then examined the position of hlePFCs associated with different levels of taxonomic annotations in the CCA space. hlePFCs containing sequences from the phylum Candidatus Marinimicrobia were found mostly in Mediterranean samples and were almost absent from polar waters (Fig. 4A). Overall, the various phyla did not show any strong associations to a particular niche in the CCA space (cf. the central positions and large standard deviation bars for the various phyla on Fig. 4A). Similarly, most of the classes' barycenters on Fig. 4B showed important standard deviation bars, with the exception of the cyanobacterial class of Prochlorales which seemed restrained to warm and oligotrophic waters (Fig. 4B). No particular order, family, or genus was found to be strongly associated with Mediterranean samples, and the genus *Polaribacter* was the only taxa strongly associated with polar samples (barycenter position on CCA2 = −1.46).

In total, 3345 hlePFCs were highly overabundant in the Mediterranean Sea (CCA1), corresponding to 8736 proteins from 193 different MAGs of 10 classes. 85 of these MAGs originated from the same assembly performed by Delmont et al.[21] on Mediterranean samples, and accounted for 5537 proteins (63.4% of the 8736 present in the zone, Supplementary Fig. 4). A similar yet less marked pattern was observed along CCA2, with 189 of the

697 (26%) proteins of hlePFCs correlated to cold and rich waters (CCA2 < −2) coming from MAGs of the Southern Ocean assembly (Supplementary Fig. 4).

Our analysis allowed us to identify environmental variables driving the abundance of functionally unannotated hlePFCs. For example, PFC #90,382 was composed of eight unannotated proteins coming from four different MAGs (3 Flavobacteriales, 1 Gammaproteobacteria), and had a strong response to high temperature (Fig. 5A), as well as other environmental variables ($R^2$ value of 0.501 for the associated random forest model). Conversely, PFCs #102,286 (two proteins coming from the same Saprospiraceae, $R^2$ value of 0.68), #210,456 (two proteins from two distinct Flavobacteriaceae, $R^2$ value of 0.83), and #233,673 (two proteins from two distinct Flavobacteriales, $R^2$ value of 0.63) were highly linked to cold temperature (Fig. 5A). PFCs #172,397, #172,465 and #26,732 were dark hlePFCs overabundant in Mediterranean samples (Fig. 5B).

Positions of all functionally and/or taxonomically unannotated hlePFCs in the CCA space, the most important drivers of their abundance according to random forest models, the nucleotidic sequences of their proteins, and their MAGs of origins are all publicly accessible (link in "Data availability").

**Robustness of the observed biogeographical patterns.** In total, 6.2% of the 233,756 PFCs that were created in this study were used in our biogeographical analysis. In section II of Supplementary notes, we provide a similar biogeographical analysis, this time including the 130,651 PFCs showing $R^2$ values above 0.25, i.e., 55.9% of the total 233,756 PFCs. A CCA based on these

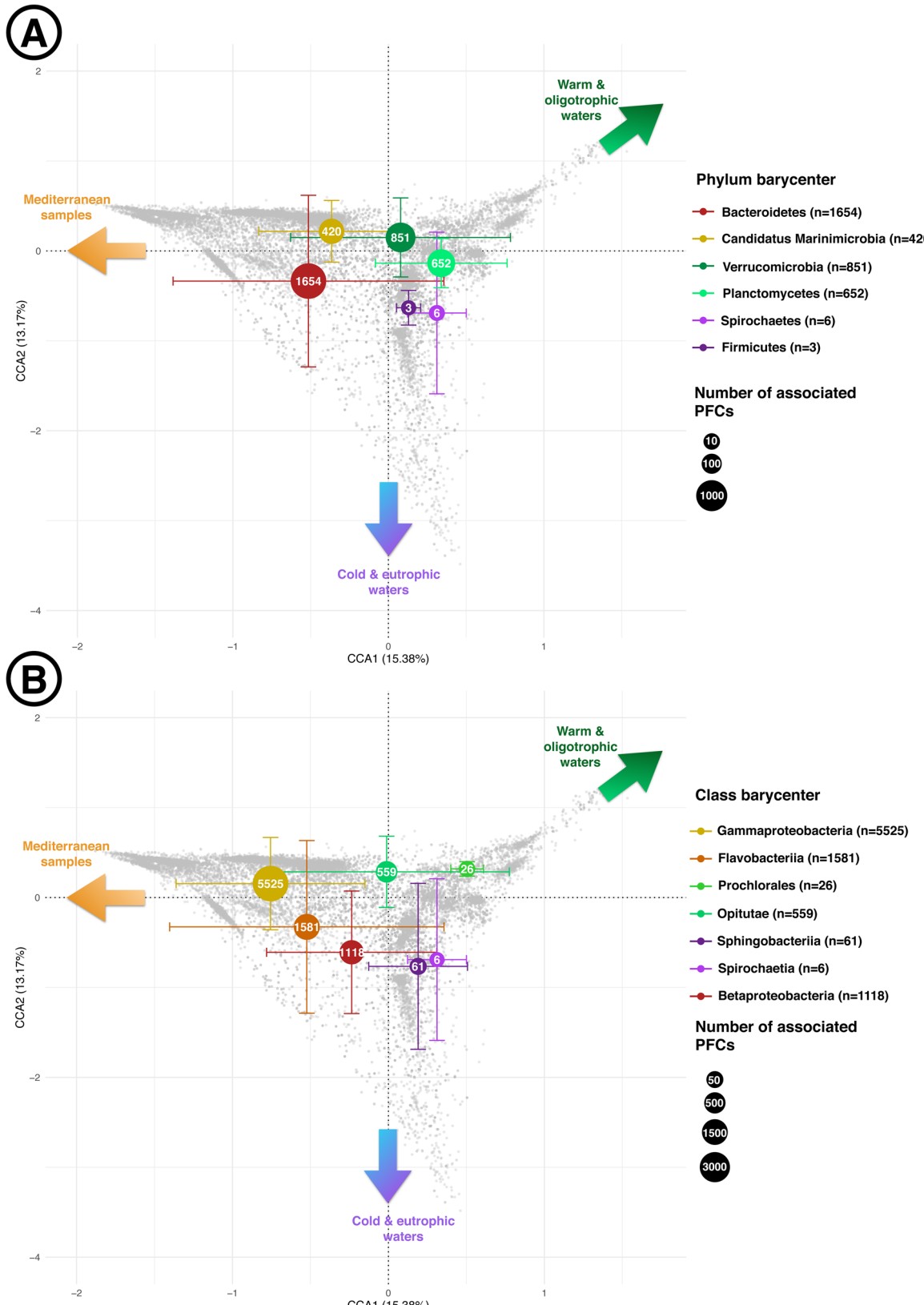

**Fig. 4 Distribution in the canonical correspondence analysis (CCA) space of the barycenters of protein functional clusters highly linked to the environment (hlePFCs) associated with particular taxa. A** Six selected phylum and (**B**) seven selected classes. These taxa were selected because they had the most peripheral barycenters' positions in the CCA space. Error bars correspond to the standard deviations of hlePFCs positions around their barycenters on CCA1 and CCA2 axes for each taxon. The size of barycenters represents the number of associated hlePFCs for each taxon, with the exact corresponding values written in white in each barycenter and in the legend. Colored arrows indicate the environmental conditions associated with the different zones of the CCA space (cf. Fig. 2).

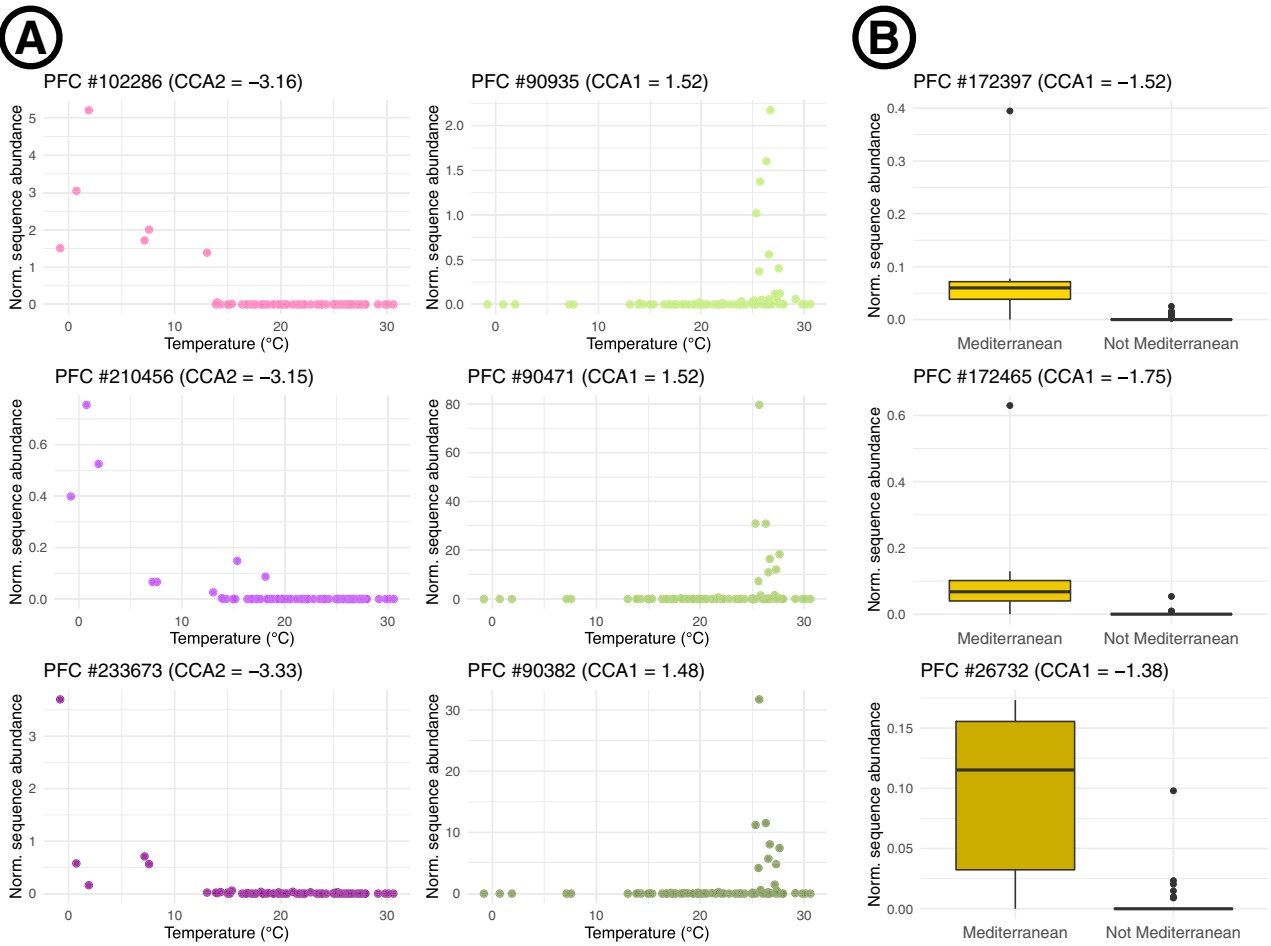

**Fig. 5 Detecting functionally unannotated protein functional clusters highly linked to environmental conditions. A** Relationships between normalized sequence abundance and temperature for six selected protein functional clusters highly linked to the environment (hlePFCs) that were only composed of functionally unannotated sequences. The three graphs on the left, in purple, correspond to the three hlePFCs functionally unannotated in both KEGG and eggNOG databases that had the lowest positions along the second axis of the canonical correspondence analysis (CCA2) (cold and nutrient-rich waters). The three graphs in the middle, in green, correspond to the three hlePFCs functionally unannotated in both KEGG and eggNOG databases that had the highest positions along CCA1 (oligotrophic and warm waters). **B** Relationships between normalized sequence abundance and location of sampling, whether in the Mediterranean Sea or not, for 3 dark hlePFCs (only functionally unannotated sequences and no taxonomic annotations under the phylum level). These 3 hlePFCs had the lowest positions among dark PFCs on the first axis of the canonical correspondence analysis (CCA2) (correlated to Mediterranean samples). Each boxplot summarizes the abundance values of its focal PFC in Mediterranean samples ($n = 6$) and non-Mediterranean samples ($n = 87$). Boxplots minima and maxima correspond to −1.5 and 1.5 times the interquartile range, while the limits of boxes correspond to the first and third quartiles. The centerline indicates the median. All points outside of the minima-maxima range are plotted.

130,651 PFCs allowed to identify two samples from the surface and deep chlorophyll maximum of station 93 as strong outliers due to their singular position along CCA1 (Supplementary Fig. 5A). The outlying nature of these two samples was explained by the strong overabundance of PFCs composed of proteins from 5 MAGs of the genus Pseudoalteromonas (Supplementary Figs. 6 and 7). The general patterns observed using the 14,585 hlePFCs (Fig. 2) were confirmed by the third and fourth dimensions of the CCA on 130,651 PFCs (Supplementary Figure 5B).

## Discussion

**The functional composition of bacterial/archaeal planktonic communities is driven by interactions between multiple environmental factors rather than by single variables**

Building statistical models including 52 environmental variables to test for their effect on PFCs abundance, we were able to quantify the impact of each variable both globally and in each individual random forest model. Our results hence give access to the most influential predictors of 228,914 PFC's abundances (see

"Data availability"), while pushing towards a consideration of other variables in addition to temperature and oxygen when studying bacterial/archaeal communities functional composition. Indeed, water temperature is commonly presented as the most influential determinant of the taxonomic and functional composition of bacterial/archaeal communities[5,23,43]. Here, we found temperature to be one of the best quantitative predictors of PFC abundance. Still, it was determined as less important than other quantitative variables like iron, carbonate, or oxygen concentrations, but also salinity or bathymetry. However, when the temperature was the most important variable in a model, it increased the accuracy of abundance predictions, and it had a strong influence on the biogeography of hlePFCs, showing how key this variable is on at least some ecosystemic functions. Among all environmental predictors, Longhurst biogeographical provinces[42] were by far the most important variable in our random forest models and were well distinguished on the CCA triplot (Fig. 2). Longhurst provinces represent homogeneous areas both in terms of physico-chemical and ecological conditions[42], and the fact that

temperature is one of the main variables used to define such provinces could explain why it was not among the most important variables in many statistical models, as the two variables could bring redundant information in regression trees. Here we thus suggest that functional composition is more impacted by interactions between multiple variables, than by one or a few variables like it has been previously suggested[5,23]. This result adds to a similar observation made by a recent global biogeographical analysis of planktonic communities, finding Longhurst biogeographical provinces to match the distribution of viruses, bacteria, and eukaryotes smaller than 20 μm[44]. Sampling depth had among the lowest impacts on our regression models, confirming the weak differences in functional composition between surface and deep chlorophyll maximum samples of picoplankton[23].

**Identifying PFCs and metabolic pathways associated with particular environmental conditions**. The identification of biogeographical provinces as best predictors of PFC's abundance can be interpreted as a consequence of the Baas Becking hypothesis "everything is everywhere, but the environment selects"[45,46], which implies that all microbes are potentially ubiquitous, but dominant taxa depend on the environmental niche. A precedent study observed this pattern at the protein level, by comparing protein families sampled in different ecosystems such as seawater, sludge water, or soils, and showing that the ecosystem type had more impact on protein families composition than geographical distance[46]. Another study based on meta-omics data found the environmentally-mediated selection to have a strong impact on the biogeography of the cosmopolitan SAR11 order Pelagibacterales[43]. Here, by identifying Longhurst biogeographical provinces to be the best predictors of PFCs abundance, we verify that environmental niches are the most determinant drivers of marine bacterial/archaeal communities functional composition. However, the fact that 44.1% of our PFCs showed poor responses to environmental conditions challenges the extent of applicability of the Baas Becking hypothesis at the protein level, at least within a single ecosystem. It could be explained by the high decoupling observed between functional diversity and taxonomic diversity among marine bacterial/archaeal communities[17]. Indeed, functional redundancy among bacteria/archaea can lead to stable functional diversity even with high taxonomic variability[47]. In our analysis, we chose to focus on changes in communities functions because it could lead to more stable abundance measures than when relying on taxonomic entities, and provide thus more valuable information on the ecosystem functioning and associated biogeochemical functions[17,23,47].

Still, 55.9% (130,651 over 233,756) of the PFCs were linked to environmental gradients, and 6.2% (14,585 over 233,756) showed strong responses to particular environmental conditions. Among these 14,585 PFCs identified as hlePFCs, we observed a clear distinction between the ones associated with polar nutrient-rich waters and those abundant in tropical nutrient-poor ones, which is coherent with classical observations in marine ecology[30,48]. Metabolic pathways like *RNA polymerase* or *Ribosome* could be associated with eutrophic conditions and colder waters (Fig. 3), potentially reflecting the higher growth potential and metabolic activity of micro-organisms in the eutrophic conditions of the polar summer[49]. No particular metabolic pathway could be associated with warm and oligotrophic waters, but proteins from a class of Cyanobacteria were overrepresented in hlePFCs abundant in these waters, which may reflect that cyanobacteria are particularly abundant in subtropical waters[50]. Biogeochemistry-related pathways such as *methane metabolism, carbon fixation in photosynthetic organisms, carbon fixation pathways in bacteria/archaea, or nitrogen metabolism* were correlated to a wide range of physico-chemical conditions

(Supplementary Fig. 3). Hence, such key biogeochemical functions seem ubiquitously present in the global ocean but can be achieved by different actors and protein families depending on the environmental conditions.

More surprisingly, we identified Mediterranean samples as clear outliers, with an important part of hlePFCs showing higher abundances in Mediterranean samples than elsewhere. These samples could not be characterized by the relative over-abundance of particular metabolic functions, and only displayed a light over-abundance of proteins from MAGs of the Candidatus Marinimicrobia phylum, which is composed of poorly known and yet uncultivable bacteria of potentially high biogeochemical impact[51]. Our strongest hypothesis to explain this pattern lies in the fact that the Mediterranean Sea is a semi-enclosed sea that experienced multiple isolations and colonization events[52]. For some pelagic species, the strait of Gibraltar constitutes a phylogeographic barrier causing genetic contrasts between Atlantic and Mediterranean populations[52,53]. Here, we identified most Atlantic samples (especially the South Atlantic ones) to be closer to Pacific samples than to Mediterranean ones in terms of hlePFCs composition, with the exception of one which came from the mouth of the strait of Gibraltar. The Mediterranean Sea was the biogeographical zone exhibiting the strongest over-abundance of locally assembled proteins, while even hlePFCs associated with Southern Ocean samples contained many proteins from Atlantic-assembled MAGs. This way, hlePFCs overabundant in Mediterranean samples shared only very few links with proteins from MAGs of other assemblies in our SSN, highlighting their functional and taxonomical originality. We then propose that the strait of Gibraltar and the Suez canal could shape the genetic and functional structure of some planktonic bacterial/archaeal populations, as it is observed in some eukaryotic species[52,53].

Finally, our analysis of PFCs showing $R^2$ values above 0.25 (Section II of supplementary notes) highlighted station 93 as a strong outlier in PFC composition. This station is the closest to the Chilean coast in our dataset and corresponds to an upwelling zone, which makes it an original station in terms of environmental conditions. It was characterized by the over-abundance of PFCs associated with the *Pseudoalteromonas* genus. *Pseudoalteromonas* is a cosmopolitan genus of Gammaproteobacteria that is known for producing biologically active molecules notably including toxins, anti-bacterial and anti-fungal agents, sometimes used as weapons against other organisms and which can be pharmaceutically relevant[54,55]. *Pseudoalteromonas* bacteria are known for being able to survive in extreme environments[56] and are not commonly associated with upwelling areas, although they are known to interact with macroalgae in such nutrient-rich coastal environments[57]. Here, we give the public access to a set of 1928 PFCs associated with the *Pseudoalteromonas* genus that was particularly abundant in a strong upwelling area, among which 127 were unannotated in both KEGG and eggNOG databases (see "Data availability"). These clusters could be related to ecological interactions between *Pseudoalteromonas* bacteria and algae, but the originality of station 93 in our dataset as a coastal and upwelling area prevents us from drawing any strong conclusions on their ecological role here. In section II of the Supplementary notes, we discuss why these PFCs were not found among hlePFCs, and highlight how the inclusion of more samples from similar coastal and/or upwelling areas would help to strengthen our results concerning these *Pseudoalteromonas* related PFCs.

**Mining the unknown to identify potential key organisms and proteins**. The main originality of our approach is its ability to take into account both annotated and unannotated sequences. It enables the identification of PFCs composed of functionally

unannotated sequences, of taxonomically unannotated sequences, and of both, leading here to the inclusion of at least 15% more proteins than methods excluding functionally unannotated sequences. By including 7834 PFCs corresponding to 20,552 protein sequences that could not be annotated under the phylum level nor to a biological function, we propose an original way to highlight the response of dark omics abundance to environmental gradients. While a previous study estimated that the inclusion of dark omics sequences could increase by up to 58% the amount of analyzed sequences[40], we provide here a pragmatic bioinformatic pipeline that helps to extend our knowledge in environmental microbiology.

It is often proposed that most of the unidentified microbial diversity could come from rare organisms, described as part of the "rare biosphere", and which are considered as diversity reservoirs able to respond rapidly to environmental changes[58,59]. Our results partly corroborate this theory (see section I of Supplementary notes), but we also found the 7834 microbial dark PFCs to be relatively overabundant in 41% of our samples. This can be explained by the fact that 72.5% of the proteins from our dark PFCs came from Candidatus Marinimicrobia and Euryarchaeota MAGs, two yet poorly studied and uncultivable phyla identified as highly abundant in the global ocean, and potentially impacting biogeochemistry[51,60,61].

Our analysis allowed us to describe the biogeography of 1347 functionally unannotated hlePFCs, which might participate in metabolic pathways involved in functional responses to peculiar environmental conditions. They included 166 dark hlePFCs, mainly related to Proteobacteria and Candidatus Marinimicrobia MAGs, the latter being associated with Mediterranean samples. In addition, more than half (52.9%) of the models associated with dark PFCs showed $R^2$ values above 0.25, and the mean $R^2$ values over the 7834 models associated with dark PFCs were similar to the one over all models. We then show that the response of dark PFCs to environmental gradients is comparable to the one of taxonomically and functionally annotated PFCs.

Our method being applicable to any set of sequences, we predict that an accumulation of similar results on multiple datasets will help identify recurrent unannotated protein clusters linked to specific environmental niches[62]. It could further help to target wet-lab studies towards the description of unknown proteins particularly adapted to specific conditions, like sub-tropical nutrient-poor waters or oxygen minimum zones[51]. However, functionally unannotated hlePFCs sometimes contained proteins from only one MAG, and in this case their response to environmental gradients could be a reflection of the global abundance of this MAG instead of a real functional level response. We then advise future wet-lab investigations to mainly select PFCs involving proteins from different MAGs. To pave the way for such further analyzes, we have provided all nucleotide sequences for each microbial dark matter PFC, as well as the statistics associated with their response to environmental gradients (see "Data availability").

## Towards more global quantitative studies of meta-omics at the function level.

Statistical models in this study were based on the abundances of each PFC in 93 metagenomic samples. For each random forest model, 75% of the samples (i.e., 70 samples) were used as a training set. Even though each model was run 10 times on 10 distinct training sets, it remains a relatively low amount of samples to do abundance predictions and extrapolations at the global ocean scale (as a way of comparison, 181 samples allowed to predict diatoms abundance from environmental data in a Chinese river[63]). Hence, machine learning models were not used to provide extrapolated predictions in this study, but to detect

PFCs hlePFCs and the main drivers of their biogeography. However, as more and more omics datasets are collected in the global ocean[23,26,64,65], we assume that similar approaches could be conducted with much more samples in the near future, which should increase models' performances. By using less than 100 samples, we were nonetheless able to obtain 14,585 models with $R^2$ values over 0.5. It highlights the potential of such quantitative approaches for predicting the abundance of key protein families in the global ocean. Moreover, our dataset was only composed of metagenomics samples, when it is hypothesized that a big part of bacterial/archaeal communities response to environmental change comes from variations in gene expression[66]. This assumption was recently disputed[23], but applying our method to metatranscriptomes in the future would allow using the environmental context to predict protein expressions instead of metagenome sequence abundances, which could help to improve the accuracy of models predictions.

In the future, biogeochemical modeling should benefit from our ability to quantify and predict biological functions using environmental and omics data[10,13,14,24,67]. Through our quantitative and data-driven analysis, we have shown one illustration of how metagenomics data can be used without a priori choices of a taxon or metabolic function. We (1) identified qualitative variables such as ocean regions and Longhurst provinces to be more informative than single quantitative variables to predict the functional composition of marine bacterial and archaeal communities, (2) identified temperature not to be a better predictor of PFC abundance in comparison to other correlated quantitative variables such as oxygen or salinity, (3) identified Mediterranean samples as outliers in terms of PFC composition, which to our knowledge had not been observed before using meta-omics data, (4) investigated dark PFCs response to environmental fluctuations, whereas the majority of studies ignore them[41]. Finally, we provide public access to PFCs along with their taxonomic and functional annotations, but also information on their relationships with particular environmental contexts, the predictability of their abundances, and their most important environmental drivers. We have then paved the way for more quantitative analysis taking advantage of the richness of global omics datasets, both at the functional and taxonomic level, which should in the long term increase our ability to better predict future global climate.

## Methods

**Samples collection and MAGs.** We focused our study on the 885 non-eukaryotic MAGs made publicly available[21]. The whole bioinformatic workflow designed to build these MAGs, as well as all the links leading to the fasta files and Anvi'o[68] profiles for each MAG can be found at http://merenlab.org/data/tara-oceans-mags/. These MAGs were built from 93 *Tara* Oceans metagenomes retrieved from 61 surface samples and 32 deep chlorophyll maximum samples collected worldwide in the global ocean, using a size filter targeting free-living microorganisms (0.2–3 μm). Original metagenomes are available under the European Bioinformatics Institute (EBI) repository with project ID ERP001736. To date, the work achieved by Delmont et al.[21]. constitutes the only database of manually refined MAGs constructed using the *Tara* Oceans project data. Automated binning efforts provided larger numbers of MAGs and focused on multiple sizes fractions[20], but are subject to higher binning errors, causing sometimes obvious contigs misplacement (as discussed here: https://bjtully.github.io/posts/2018/10/re-visiting-tmed-mags/). Further information on the MAGs' genomic features, such as their completion or GC content, can be found in Supplementary Table 5 of Delmont et al.[21].

**Gene detection and quantification.** Prodigal v2.6.3[69] was run to retrieve the nucleotide and protein sequences of each detected gene for each of the 885 MAGs. By concatenation, one nucleotide and one protein fasta files were created, containing each in total 1,914,171 sequences. The nucleotide sequences were then used for the mapping and quantification step (hereafter developed) whereas the protein sequences were used for building the SSN (cf. next paragraph).

The nucleotide file was used as an index to quantify the MAGs' genes abundance in the 93 metagenomes used by Delmont et al. for the MAGs binning process[21]. For this, we mapped metagenome reads to the MAGs gene catalog using the *quant*

function from Salmon v.0.11.3[70] in quasi-mapping mode, with the following parameters "–libType A–meta–incompatPrior 0.0–seqBias–gcBias–biasSpeedSamp". To normalize the obtained read counts, we divided them by the gene length, and by the total of sequenced reads per sample, then multiplied them by 10e9. The obtained value is analogous to RNA-seq transcripts per million value (TPM), except that TPM calculation is based on the total amount of reads that mapped to the transcripts index, while we used here the total amount of reads that have been sequenced in each sample (e.g., mapped + unmapped). In fact, the underlying assumption behind TPM and other RNA-seq orientated normalizations is that all compared samples should come from similar tissues, hence displaying a comparable number of mapped reads, which is incompatible with environmental metagenomics. Indeed, *Tara Oceans* samples contain variable quantities of biological matter coming from different sampling in the global ocean, leading them to have very variable amounts of total sequenced and mapped reads. Typically, if a sample has a high total number of sequenced reads but a low number of mapped reads, it will still display high abundance values for the few mapping reads when using the classic TPM normalization, while it would not be the case with our method.

**Building an SSN from 885 bacterial/archaeal MAGs.** An SSN is a graph object in which vertices correspond to sequences and edges represent the similarity and coverage between pairs of sequences[31–35]. Diamond v0.8.22[71] was used in blastp mode to compute the percentage of similarity between every pair of proteins detected in the MAGs, using options "−e 1e−3 −p 30–sensitive". An SSN was built with the diamond output using 80% identity and 80% coverage threshold. This coverage threshold is commonly used in SSN studies[33,35,72] and we also tested 4 other similarity thresholds: 70%, 75%, 85%, and 90%. We selected the intermediary 80% identity threshold to minimize the amount of singletons while maximizing the functional homogeneity between linked proteins.

**Extracting, annotating, and quantifying PFCs in the SSN.** An SSN is made of singletons (vertice or sequence without any homology with other sequences) and CCs (subgraphs composed of at least two vertices disconnected from the rest of the network). In our case, a CC corresponds to a group of at least two protein sequences that are linked together (directly or via neighbors), and that have no link with other groups of sequences in the SSN. We assume that the proteins contained in a CC potentially share a similar molecular function[31–33,72]. The term "protein family" is often used to describe such clusters of homologous proteins, but as this term is usually used to deal with evolutionary relationships, we here prefer the use of PFC.

Our SSN was composed of 233,756 PFCs, including 757,457 proteins (i.e., 1,156,714 singletons were excluded from the analysis). These proteins were functionally annotated using eggNOG mapper v4.5.1[38,39] and KofamScan v1.2.0[37]. EggNOG emapper was run using the diamond mode and the–no_annot flag. It produced a table containing seed orthologous sequences for 677,684 of our proteins (89.5%), the rest of them not being similar enough from any sequence in the eggNOG database. The annotation phase was then launched on these 677,684 proteins, using the seed orthologous sequences table as input to the emapper function, and the–annotate_hits_table flag. We obtain an annotation table with GO IDs, KEGG IDs, and eggNOG descriptions. KoFamScan was launched with default options and -mapper flag. The KEGG API was then used to retrieve KEGG pathways ID and descriptions for each KEGG ID identified by KoFamScan in our protein catalog. To assess for the functional homogeneity in our PFCs, we computed a homogeneity score $F_{hom}$:

$$N_{annot} > 1 \Rightarrow F_{hom} = 1 - \frac{N_{annot}}{N_{prot}} \tag{1}$$

$$N_{annot} = 1 \Rightarrow F_{hom} = 1 \tag{2}$$

With $N_{annot}$ the number of unique annotation terms found in the PFC (either KEGG IDs or eggNOG terms), and $N_{prot}$ the number of proteins in the PFC.

As multiple eggNOG terms can exist for similar functions (e.g., "UBA-ThiF-type NAD FAD-binding protein" and "UBA-THIF-type NAD FAD-binding"), they can lead to artifactually low homogeneity scores. For this reason, PFCs with low homogeneity scores obtained with the EggNOG database were tagged as poorly homogeneous but were kept in the analysis.

Statistics on functionally unannotated PFCs presented in Tables 1 and 2 include both (1) query sequences that did not match to any reference in public databases, and (2) query sequences that match to one or multiple references in public databases but could not yet be associated to any biological function.

To assess taxonomic diversity in our PFCs, we used the taxonomic annotation of the 885 MAGs provided by Delmont et al.[21]. This taxonomic annotation was inferred through 43 single-copy core genes through the combined use of CheckM[73], RAST[74], and manual BLAST searches (see[21] for further details).

We computed a mean abundance for each PFC in each of the 93 metagenomes, using relative protein abundances (see *Gene detection and quantification*). We obtained an abundance table composed of 233,756 rows, corresponding to PFCs, and 93 columns, corresponding to the 93 *Tara Oceans* metagenomes used in the study.

**Environmental dataset.** For each of the 93 *Tara Oceans* metagenomes, we retrieved the environmental context from Faure et al. (https://figshare.com/articles/Data_MixoBioGeo_Faure_et_al_2018/6715754)[30]. To complete this environmental dataset, we added 10 climatology variables retrieved from the World Ocean Atlas:[75] temperature, salinity, density, conductivity, dissolved oxygen, percent oxygen saturation, apparent oxygen utilization, silicate, phosphate, and nitrate. Finally, we added the upper limit of the size fraction as an environmental variable, as it varied across samples. For temperature, salinity, and conductivity we retrieved the mean and the mean seasonal anomaly at each sampling point (precision of 1°) over the 2005–2012 period. Only the mean was retrieved for density. For the 6 other variables, we retrieved the mean and the mean seasonal anomaly at each sampling point (precision of 1°) over all available years. In total, we obtained 74 environmental variables, which we reduced to 51 by getting rid of near-zero variance variables and too highly correlated ones, using options "nzv" and "corr" from the preProcess function of the *caret* package v6.0[76] in R v3.5.3[77]. A detailed description of these variables is available in Supplementary Data 2. We then scaled and centered the 51 selected environmental variables, and used a k-nearest neighbors approach to replace NA values (6.6% of the data) by the mean of the concerned variable in the five nearest samples in terms of global environmental profile (knnImpute option from caret's preProcess function[76]).

**Identification of PFCs varying along environmental gradients.** Among the 233,756 PFCs, we detected 4842 (2,1%) clusters with near-zero variance using caret preProcess function[76] with default parameters, i.e., they had less than 10% of abundance values across all samples that were distinct, and a ratio between the most common abundance value and the second most common one that was higher than 95–5. These clusters were removed from further statistical analysis. We built a random forest regression model for each of the remaining 228,914 PFCs, using the environmental variables as predictors of cluster relative abundance. To suppress eventual biases linked to over/underfitting due to training set selection, each model was launched 10 times using 10 different training sets built using 75% of the 93 samples available. For each iteration, i.e., for each pair of the training set and PFC, a random forest regression model was trained using a fivefold cross-validation process, and the number of randomly tested predictors at each split was optimized between a minimum of 5 and a maximum of 9 (the default value being 7, the floored square root of the number of environmental variables), while the number of trees was fixed to 500. The model minimizing the root mean square error (RMSE) was selected for each iteration and used to compute the mean prediction error over the 10 iterations, as well as the mean $R^2$, and the mean rank of importance in the model for each environmental predictor. For each iteration, the selected model was also used to produce predictions over the test set, and the mean $R^2$ of these predictions was computed over the 10 iterations. This way, we have for each PFC an $R^2$ based on the out-of-bag samples of the cross-validation process, and an $R^2$ based on predictions over the test set (Fig. 1B). The mean $R^2$ obtained from the out-of-bag samples was used to discriminate PFCs following significant environmental gradients from the ones showing no response to the environmental context. Specifically, we considered every PFCs associated with a model with a mean $R^2$ value over the arbitrary threshold of 0.5 to be hlePFCs, and over 0.25 to be linked to environmental gradients. Different thresholds ranging from 0.25 to 0.75 were tried for the definition of hlePFCs, thresholds higher than 0.5 tended to select too few PFCs, and mainly the ones overabundant in the Mediterranean Sea, while too low thresholds tended to diminish the $R^2$ value and readability of the CCA (*cf* next section). All random forest models were launched using the *rf* function of the randomForest v4.6 R package[78] through the *train* function of the Caret package[76].

**Biogeography of PFCs linked to environmental gradients.** We used a CCA to describe in a more integrated way the relationships between PFCs and environmental variables. The CCA used the relative abundance table of all PFCs linked to environmental gradients (mean $R^2$ of random forest regressions > 0.5) as response variables, and 17 selected environmental variables as explanatory variables: biogeographical province, ocean region, season moment (i.e., early/middle/late), temperature, depth, depth of the euphotic zone, conductivity seasonal anomaly, sea surface temperature gradient, moon phase proportion, depth of the $O_2$ minimum, calcite saturation state, fluorescence, $NO_3$ at 5 m, chlorophyll a, total alkalinity, salinity, iron at 5 m. The 17 environmental variables were selected through a backward and forward stepwise selection based on the AIC criterion[79].

Using positions of PFCs in the two first dimensions of the CCA space (29.09% of variance), we computed a barycenter position for each metabolic pathway detected among hlePFCs (Fig. 3). Similarly, we computed barycenters for phyla, classes, and genomic assemblies in the CCA space (Fig. 4, Supplementary Fig. 3). Finally, convex hulls englobing all PFCs associated with a pathway were drawn for a selection of pathways corresponding to (1) pathways linked to inter-organisms interactions, (2) pathways associated to a priori selected biogeochemical functions, and (3) pathways composed of only unknown sequences (Supplementary Fig. 3).

The exact same methods were applied to compute the biogeographical analysis of the PFCs associated with the models with $R^2$ values above 0.25 (see Section II of Supplementary materials).

**Reporting summary**. Further information on research design is available in the Nature Research Reporting Summary linked to this article.

## Data availability

Instructions on how to build or download the MAGs and metagenomes used in this study are available at http://merenlab.org/data/tara-oceans-mags/. Tools and databases used for functional annotations are available at http://eggnog-mapper.embl.de/ and https://www.genome.jp/tools/kofamkoala/. All other data used in this study are available at 10.6084/m9.figshare.12030795, including fasta files containing nucleotide sequences of all proteins in PFCs, hlePFCs, dark PFCs, PFCs associated with Station 93, and PFCs associated with Station 93 linked with Pseudoalteromonas MAGs. In this figshare repository, we also provide summary tables including all PFC and random forest associated statistics (e.g., all homogeneity and unknown scores, $R^2$ values, variables importances) for each PFCs, hlePFCs, and dark PFCs. Finally, we offer tables at the single protein level showing the PFC ID, taxonomic and functional annotations, and nucleotide sequences of each protein in PFCs, hlePFCs, and dark PFCs.

## Code availability

All bash, perl, and R codes necessary to reproduce our analysis are available at https://github.com/EmileFaure/MAGsProteinFunctionalClusters[80].

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

## Acknowledgements

We would like to particularly thank Loïs Maignien, Olivier Aumont, Eric Pelletier, Christian Tamburini, Sébastien Monchy, Thomas Mock, and Ingrid Obernosterer for the insightful discussions concerning this study. We also thank the Meren Lab (https://merenlab.org/) and all the people involved in *Tara Oceans* for producing the data we used and making them publicly available. We would also like to thank all participants of the GOBITMAP and GREENOCEAN workshops for their useful advice. Finally, EF would like to thank Raphaël Berthier and Jean-Olivier Irisson for their advice concerning machine learning, as well as François Duchenne, Elise Kerdoncuff, Benoît Pérez and Eric Bapteste for their helpful comments. This work was funded mainly by our salary as French State agents and therefore by French taxpayers' taxes. EF acknowledges a 3-year Ph.D. grant from the "Interface Pour le Vivant" (IPV) doctoral program of Sorbonne Université. SDA acknowledges the CNRS for her two sabbatical years as visiting researcher at ISYEB in 2018-2020. LB acknowledges the Institut Universitaire de France for her 5-year nomination as Junior Member (2020-2025). Additional support was provided by the Institut des Sciences du Calcul et des Données (ISCD) of Sorbonne Université through the support of the sponsored junior team FORMAL (From ObseR-ving to Modeling oceAn Life). EF acknowledges the financial help of the Korean Institute of Ocean Science and Technology to attend the IMBeR 2019 conference, which led to very helpful discussions concerning this work.

## Author contributions

E.F., S.D.A. and L.B. conceived the study. E.F. processed and analyzed the data, with inputs from S.D.A. and L.B. E.F., S.D.A. and L.B. wrote the manuscript.

## Competing interests

The authors declare no competing interests.
