## [Peer Review File · Nature Communications]

Reviewer comments, first round

Reviewer #1 (Remarks to the Author):

In the manuscript entitled "Towards omics-based predictions of planktonic functional composition from environmental data", Emile Faure, Sakina-Dorothee Ayata and Lucie Bittner tackle an interesting question: can the abundance of protein clusters of known and unknown function derived from genome-resolved metagenomics be "predicted from environmental data in the oceanic ecosystem"?

In the study, authors used 51 quantitative and qualitative environmental variables in order to identify, from a set of manually curated marine MAGs (nearly two million proteins), protein clusters that are sensitive to environmental gradients. Notably, authors applied machine learning to link the distribution of more than 200,000 protein clusters across oceanic regions to biogeographic provinces and environmental data, using 93 metagenomes from TARA Oceans. As the authors suggest, the methodology outlined in this study could be applied to other ecosystems and genomic databases. Finally, data produced in this study has been made available.

The manuscript is clear, concise and well written. The methodology is sound. For example, functional homogeneity of the protein clusters was carefully considered, and the link between clusters and taxonomy well described. Identification of the hlePFCs, and their subsequent description, is certainly interesting. In my view, it successfully demonstrates the value of this bioinformatics approach to gain insights into the functioning of plankton at large-scale, with "no a priori choice of specific functions or taxa", as rightfully emphasized by the authors in the introduction. Most importantly, authors successfully linked protein clusters, environmental variables, functions and taxonomy for a wide range of mostly bacterial populations abundant in the sunlit ocean, exposing among other insights some protein clusters of unknown functions and poorly characterized taxonomy enriched in the Mediterranean Sea.

The only concern I have is with regard to one variable authors might have overlooked to some extent: the size fraction differences between legs of the TARA Oceans expeditions. Indeed, the expeditions experienced a methodological change between oceanic regions regarding the 93 metagenomes considered here: 0.2-1.6 micron for Mediterranean Sea (plus Atlantic station on the way to the Mediterranean Sea), Red Sea and Indian Ocean, and 0.2-3 micron for the Atlantic, Pacific and Southern Ocean in subsequent expeditions. This not only impacted the characterization of MAGs (Delmont et al. 2018), but it also likely impacts to some degree the distribution of protein clusters the authors relied on here. For example, one could imagine that protein clusters from relatively large bacterial cells will be less likely to be detected in the Mediterranean Sea, Red Sea and Indian Ocean. My concern is that the clustering of samples in figure 2 could be governed, at least to some degree, by this methodological bias of the TARA Oceans expeditions. I would like to emphasize here that the clear differences between Indian Ocean and Mediterranean Sea suggest most insights are biologically relevant. Yet, authors need to consider this issue, and if possible demonstrate that the main observations presented in this study are not impacted by size fraction incoherence between metagenomes. If this is not possible, then the shift in size fractions between oceanic regions will have to be described as a limitation in the analysis.

Some relatively important points:

I would like clarification regarding text in Ln 183-189 and their link to Figure 1B. I have difficulties seeing coherence between the two. Assuming I do not correctly read the figure, please explain to me why biogeographical province is discussed as the most important factor, yet is at the very left in the figure, with a boxplot mean of <5%.

Please make sure that MAGs containing the 24 hlePFCs of unknown functions do not have a drop in completion compared to other MAGs in the set. This is because some MAGs with >2Mbp and very low completion were initially kept in the collection of MAGs, however their quality is now disputable (Ln 221). Those MAGs are being removed from new databases under development.

Smaller points:

The term of "microbial dark matter" has been extensively criticized given the poor analogy between dark matter in physics (a truly remarkable state of matter we can only poorly interact with), and microbial populations yet to be cultured in a petri dish. Critically, no remarkable differences exist between populations in culture and those we have yet to isolate in the lab. Thank you for considering using a different terminology.

Please consider avoiding the term of "prokaryotes".

Ln 173-177: Please explain the rationale behind those decisions.

Figure 1A: could you consider plotting those numerous dots as a curve with R2 for the x-axis and quantity for the y-axis? It might improve readability of the data.

Ln 360-363 and/or Ln 410: please consider introducing a recent study on amino acid diversification of SAR11 across oceans given its content (<https://elifesciences.org/articles/46497>).

Ln 385: Cyanobacteria is known to be absent from polar regions. Thus it is not simply about nutrient-poor waters, but more about a latitudinal shift, in my view.

Ln 489: please add a reference for *anvi'o*.

Tom O. Delmont

Reviewer #2 (Remarks to the Author):

Summary:

This manuscript identified the best predictors of the distribution of protein functional cluster's abundance in marine prokaryotes, relying on previously compiled 885 marine prokaryotic metagenome-assembled genomes (MAGs) and Tara Ocean's sequencing data. The authors applied multiple statistical approaches including the sequence similarity network to determine the protein functional clusters (233,756), a machine learning technique (random forest) to pinpoint the best predictor (biogeographic provinces) and a canonical correspondence analysis to link protein functional clusters with environmental gradients. Overall, it provides an avenue/direction for future analyses of meta-omics data in quantitative predictions of functional groups using environmental variables.

The major comments on the manuscript are:

1. need to better highlight the novelty of this manuscript in comparison to other papers that have also extensively evaluated the patterns and drivers of microbial composition and function in the global ocean (e.g. Sunagawa et al., 2015; Salazar et al., 2019; Righetti et al., 2019), which presented some similar results. For example, temperature has previously been identified as a good predictor of marine microbial community composition or function at gene/transcript level, and is now found at the protein level in this study. It would be important to show the additional critical results from this work compared to the previous study that has done any pre-selected choices of particular genes or functions (e.g. Salazar et al., 2019). However, it is not very clear now.

2. predictability of protein functional clusters. Only ~1% of the detected protein functional clusters (2,444 out of 233,756) are predictable using the criterion in the study, questioning the applicability of machine learning and the subsequent major analyses based on these limited 2,444

protein functional clusters (Figure 2, 3 and 4). In this case, a substantial amount of data/information is ignored. For example, Mediterranean Sea was categorized as an outlier in these 2,444 protein functional clusters by the canonical correspondence analysis. But Mediterranean Sea may drive a large portion of the variation in PFCs (Figure 2, 3 and 4). Will it be worthwhile to re-do some of the analyses after removing data from the Mediterranean Sea and compare with current results? Besides, will it be worthwhile to conduct a canonical correspondence analysis on all of the 233,756 protein functional clusters or 44,653 protein functional clusters ($R^2 > 0.25$ in random forest model)?

3. clarity of the method and results description. This is really a comprehensive analysis on the proteins retrieved from 885 MAGs. Starting from 1,914,171 proteins to 233,756 connected components and then to 2,444 "hlePFCs", there are so many numbers and percentages in Results section in addition to Tables 1 and 2. It may be useful to have a flowchart to illustrate how to produce the different group of data (size of the data) and the exact analyses done in each step to these data. For example, 233,756 connected components were selected based on sequence similarity network and then 228,914 clusters were chosen due to the non-zero variance abundance profiles.

Here are the detailed comments:

1. Line 1: "predictions" used in the title may not be appropriate. This study mainly focuses on the correlation analysis between planktonic functions and environmental data. It does not provide any further predictions at the regional or global scale.
2. Line 129: Protein functional cluster (PFCs) instead of in line 131
3. Line 131: "numeroted" to "numbered" or "enumerated"
4. Line 184: Since Longhurst biogeographical provinces were identified as the best predictor, it may be useful to show the Longhurst biogeography map with sampling locations on top of it in the supplementary. It would help to visualize the Figure 2 legend that only shows the acronyms of these provinces.
5. Figure 1. It would be nice to show at least one example of the random forest model (observed versus predicted abundances in the training and test datasets) and explain the negative R^2 from a large number of models. Try to keep the notations of environmental variable consistent in Figure 1B and in the Table S2. For example, temperature is described in the text while temperature anomaly is shown in the x axis of Figure 1B.
6. Lines 199 to 204, could you explain why only a small fraction of the exact same metabolic pathways are predictable?
7. Line 208: "was divided by two" to "was halved"
8. Lines 242-250: Instead of categorizing Mediterranean Sea as the exception, isn't Mediterranean Sea explaining a large fraction of the variation of hlePFCs (see a large number of hlePFCs are located in the $CCA_2 > 2$)? It would be interesting to do all the analyses and see what the results look like after removing data from the Mediterranean Sea. How many hlePFCs will exist and how will the CCA plot look like?
9. Figure 2, it would be interesting to color-code the hlePFCs with R^2 values and see which dimension presents hlePFCs with high R^2 values.
10. Line 300, Atlantic North-West is not shown in Figure S3.
11. Lines 338-339, O₂ doesn't show a higher rank than temperature in Figure 2B.
12. Lines 366-368 and line 19.5% and 1.1%, it would be nice to note how these numbers are derived, e.g. (2,444/233,756).
13. Lines 387-395, they are still descriptive not explanatory. For example, why methane metabolism is one of the most selected pathways in hlePFCs while not the nitrogen metabolism? It would be good to provide some discussions.
14. Lines 396-397, again it would be interesting to run some analysis after removing the data from Mediterranean Sea. This helps to double-check whether Mediterranean Sea samples are only minor outliers or are actually driving the global pattern.
15. Line 416 and the following section, it is an originality that this study considers both annotated and unannotated sequences. But it would be good to provide some additional valuable information from this discussion in addition to suggestions for future work. Only 0.3% of the microbial dark matter (24 out of 7834) is selected as hlePFCs. What about the other unpredictable microbial dark

matter?

Methods

1. Line 486, as mentioned above, it might be clearer to read the Figure 2 if showing the sampling/data locations on top of the Longhurst provinces in a supplementary figure.
2. Lines 606-612, how is 75% of the sample selected for each training datasets? Did you have a chance to look at the results of the out of bag samples or validate the model in the test dataset to evaluate the model performance?
3. Line 643-644, it would be good to provide a text description for each file in this link <https://figshare.com/s/b33fc72a62db44b7192f>.

References:

- Righetti, D., Vogt, M., Gruber, N., Psomas, A., & Zimmermann, N. E. (2019). Global pattern of phytoplankton diversity driven by temperature and environmental variability. *Science advances*, 5(5), eaau6253.
- Salazar, G., Paoli, L., Alberti, A., Huerta-Cepas, J., Ruscheweyh, H. J., Cuenca, M., ... & Gregory, A. C. (2019). Gene expression changes and community turnover differentially shape the global ocean metatranscriptome. *Cell*, 179(5), 1068-1083.
- Sunagawa, S., Coelho, L. P., Chaffron, S., Kultima, J. R., Labadie, K., Salazar, G., ... & Cornejo-Castillo, F. M. (2015). Structure and function of the global ocean microbiome. *Science*, 348(6237), 1261359.

We would like to deeply thank both reviewers for their insightful comments which allowed us to significantly improve our work.

Our detailed answers to the reviewers are written in green in the text below. The line numbers indicated in our responses correspond to those of the manuscript **with track-changes**.

Based on the comments from both reviewers and using some help and advice from machine learning experts, we applied 3 major changes to our statistical approach. These 3 changes are briefly described here before going into more details in our point-by-point answers:

- (1) We now include the size fraction as a predictor in our machine learning pipeline, and as an explanatory variable in our canonical correspondence analysis.
- (2) To improve model training and optimization, a 5-fold cross validation step was added at each iteration, meaning that for each protein functional cluster, 10 cross-validated models were built using 10 distinct training sets. We also added a step of optimization of the “mtry” parameter, corresponding to the number of variables tested at each split of the regression tree. After each training process (*i.e.* cross-validation and optimization), the model minimizing the root mean square error was selected as the best, thus 10 trained models were selected for each protein functional cluster (one for each training set).
- (3) To avoid negative R^2 values and facilitate the inclusion of more protein functional clusters in our biogeographical analysis, we now define R^2 as the squared correlation between observations and predictions, as it is commonly done in biology.

The new pipeline is described in the manuscript in lines 1191 to 1212.

Reviewer #1 (Remarks to the Author):

In the manuscript entitled “Towards omics-based predictions of planktonic functional composition from environmental data”, Emile Faure, Sakina-Dorothee Ayata and Lucie Bittner tackle an interesting question: can the abundance of protein clusters of known and unknown function derived from genome-resolved metagenomics be “predicted from environmental data in the oceanic ecosystem”?

In the study, authors used 51 quantitative and qualitative environmental variables in order to identify, from a set of manually curated marine MAGs (nearly two million proteins), protein clusters that are sensitive to environmental gradients. Notably, authors applied machine learning to link the distribution of more than 200,000 protein clusters across oceanic regions to biogeographic provinces and environmental data, using 93 metagenomes from TARA Oceans. As the authors suggest, the methodology outlined in this study could be applied to other ecosystems and genomic databases. Finally, data produced in this study has been made available.

The manuscript is clear, concise and well written. The methodology is sound. For example, functional homogeneity of the protein clusters was carefully considered, and the link between clusters and taxonomy well described. Identification of the hlePFCs, and their subsequent description, is certainly interesting. In my view, it successfully demonstrates the value of this bioinformatics approach to gain insights into the functioning of plankton at large-scale, with “no a priori choice of specific functions or taxa”, as rightfully emphasized by the authors in the introduction. Most importantly, authors successfully linked protein clusters, environmental variables, functions and taxonomy for a wide range of mostly bacterial populations abundant in the sunlit ocean, exposing among other insights some protein clusters of unknown functions and poorly characterized taxonomy enriched in the Mediterranean Sea.

The only concern I have is with regard to one variable authors might have overlooked to some extent: the size fraction differences between legs of the TARA Oceans expeditions. Indeed, the expeditions experienced a methodological change between oceanic regions regarding the 93 metagenomes considered here: 0.2-1.6 micron for Mediterranean Sea (plus Atlantic station on the way to the Mediterranean Sea), Red Sea and Indian Ocean, and 0.2-3 micron for the Atlantic, Pacific and Southern Ocean in subsequent expeditions. This not only impacted the characterization of MAGs (Delmont et al. 2018), but it also likely impacts to some degree the distribution of protein clusters the authors relied on here. For example, one could imagine that protein clusters from relatively large bacterial cells will be less likely to be detected in the Mediterranean Sea, Red Sea and Indian Ocean. My concern is that the clustering of samples in figure 2 could be governed, at least to some degree, by this methodological bias of the TARA Oceans expeditions. I would like to emphasize here that the clear differences between Indian Ocean and Mediterranean Sea suggest most insights are biologically relevant. Yet, authors need to consider this issue, and if possible demonstrate that the main observations presented in this study are not impacted by size fraction incoherence between metagenomes. If this is not possible, then the shift in size fractions between oceanic regions will have to be described as a limitation in the analysis.

Response: We thank the reviewer for bringing up this important point that we did not address in the first version of our manuscript. The size fraction associated with each sample has now been added as a categorical variable in the environmental meta-dataset used in the study, and its potential impact was hence measured and taken into account both through the machine learning approach and the biogeographical study. As exposed in the new Figure 1C, the impact of size fraction on the abundance of Protein Functional Clusters (PFCs) was extremely limited.

Some relatively important points:

I would like clarification regarding text in Ln 183-189 and their link to Figure 1B. I have difficulties seeing coherence between the two. Assuming I do not correctly read the figure, please explain to me why biogeographical province is discussed as the most important factor, yet is at the very left in the figure, with a boxplot mean of <5%.

Response: Figure 1B becomes Figure 1C in the new version of the manuscript. This figure shows boxplots of the ranks of importance, and not percentages of importance. The fact that the mean rank of biogeographical provinces is below 5 shows that among all models, biogeographical provinces were on average the 5th most important variable. We highlighted in the legend that a lower rank indicates a higher importance in models, we hope it to be clearer now.

Please make sure that MAGs containing the 24 hlePFCs of unknown functions do not have a drop in completion compared to other MAGs in the set. This is because some MAGs with >2Mbp and very low completion were initially kept in the collection of MAGs, however their quality is now disputable (Ln 221). Those MAGs are being removed from new databases under development.

Response: We would like to thank Reviewer #1 for highlighting this potential bias. Among the original 957 MAGs published in Delmont et al. (2018), 238 had a completion below 70%, and 157 a completion below 50% (completion estimates from column *Anvio-Completion 4databases* in Table S3 of Delmont et al., 2018). In the revised manuscript, thanks to the modifications in our machine learning pipeline evoked earlier, we raise the number of dark hlePFCs from 24 to 166. These 166 PFCs are composed of proteins coming from 51 distinct MAGs. Among these 51 MAGs, 4 contributed to 69.2% of the proteins from the 166 dark hlePFCs: TARA_PSW_MAG_00045, TARA_MED_MAG_00105, TARA_RED_MAG_00051 and TARA_IOS_MAG_00051. Their completions were 93.89%, 73.76%, 86.47% and 75.2%, respectively. Among the 47 remaining MAGs, 5 had a completion below 70% and none had a completion below 50%. Hence, the association

between dark PFCs and the environmental context was not related to a drop in their associated MAGs' completion. A short sentence was added on this subject at lines 295-296.

Smaller points:

The term of "microbial dark matter" has been extensively criticized given the poor analogy between dark matter in physics (a truly remarkable state of matter we can only poorly interact with), and microbial populations yet to be cultured in a petri dish. Critically, no remarkable differences exist between populations in culture and those we have yet to isolate in the lab. Thank you for considering using a different terminology.

Response: The term of 'microbial dark matter' was replaced in the text by 'dark' omics, 'dark' PFCs or 'dark' hlePFCs.

Please consider avoiding the term of "prokaryotes".

Response: We understand the reviewer's point, and thus modified accordingly our manuscript.

Ln 173-177: Please explain the rationale behind those decisions.

Response: Original lines were: *"To identify the PFCs that responded the most to environmental gradients, we first selected the 228,914 clusters with non-zero variance abundance profiles (i.e. at least 10% of distinct abundance values across all samples, and less than a 95 to 5 ratio between the most and the second most observed abundance value)."* We now have clarified our intention to avoid the creation of constant or near constant training and/or test sets, which can not be used in random forest models. The updated text in the results section (lines 205-208) is: *"To identify the PFCs that responded the most to environmental gradients, we first selected the 228,914 clusters with non-zero variance abundance profiles (i.e. at least 10% of distinct abundance values across all samples, and less than a 95 to 5 ratio between the most and the second most observed abundance value, please see Methods for more details), to avoid the creation of constant or near-constant training and/or test sets"*. In the methods we specified that the parameters used are the default ones of the PreProcess function from the Caret R package.

Figure 1A: could you consider plotting those numerous dots as a curve with R2 for the x-axis and quantity for the y-axis? It might improve readability of the data.

Response: We would like to thank the reviewer for proposing this, the revised manuscript includes a new version of Figure 1A in which R squared values are represented as a density curve, improving the readability of the data.

Ln 360-363 and/or Ln 410: please consider introducing a recent study on amino acid diversification of SAR11 across oceans given its content (<https://elifesciences.org/articles/46497>).

Response: We have now added a sentence citing this study: *"Another study based on meta-omics data found environmentally-mediated selection to have a strong impact on the biogeography of the cosmopolitan SAR11 order Pelagibacterales [40]"* (lines 796-798).

Ln 385: Cyanobacteria is known to be absent from polar regions. Thus it is not simply about nutrient-poor waters, but more about a latitudinal shift, in my view.

Response: The sentence has been changed to *"No particular metabolic pathway could be associated with warm and oligotrophic waters, but proteins from a class of Cyanobacteria were overrepresented*

in hlePFCs abundant in these waters, which may reflect that cyanobacteria are particularly abundant in subtropical waters [47]" (lines 832-835).

Ln 489: please add a reference for anvi'o.

Response: Done, we apologize for not including it in the previous version.

Reviewer #2 (Remarks to the Author):

Summary:

This manuscript identified the best predictors of the distribution of protein functional cluster's abundance in marine prokaryotes, relying on previously compiled 885 marine prokaryotic metagenome-assembled genomes (MAGs) and Tara Ocean's sequencing data. The authors applied multiple statistical approaches including the sequence similarity network to determine the protein functional clusters (233,756), a machine learning technique (random forest) to pinpoint the best predictor (biogeographic provinces) and a canonical correspondence analysis to link protein functional clusters with environmental gradients. Overall, it provides an avenue/direction for future analyses of meta-omics data in quantitative predictions of functional groups using environmental variables.

The major comments on the manuscript are:

1. need to better highlight the novelty of this manuscript in comparison to other papers that have also extensively evaluated the patterns and drivers of microbial composition and function in the global ocean (e.g. Sunagawa et al., 2015; Salazar et al., 2019; Righetti et al., 2019), which presented some similar results. For example, temperature has previously been identified as a good predictor of marine microbial community composition or function at gene/transcript level, and is now found at the protein level in this study. It would be important to show the additional critical results from this work compared to the previous study that has done any pre-selected choices of particular genes or functions (e.g. Salazar et al., 2019). However, it is not very clear now.

Response: We thank the reviewer for encouraging us to highlight more strongly the originality of our study. To our knowledge, our study is the first to investigate patterns and drivers of microbial composition and function in the global ocean by focusing on inclusive protein functional clusters without any a priori on function or taxonomy. This allowed us to obtain original results compared to studies such as Sunagawa et al., 2015, Salazar et al., 2019 or Righetti et al., 2019: (1) we identify temperature not to be a better predictor of protein functional cluster (PFC) abundance in comparison to other correlated quantitative variables such as oxygen or salinity (Figure 1C), (2) we identify qualitative variables such as ocean regions and Longhurst provinces to be more informative than single quantitative variables (Figure 1C), (3) we identify Mediterranean samples as outliers in terms of PFC composition, when these samples have not yet been identified as outliers in other *Tara Oceans* studies to our knowledge, (4) we provide public access to PFCs based on their relationship with the environmental context, when all existing studies only provide access to such clusters based on functional annotations, (5) among these clusters identified as linked to particular environmental conditions (hlePFCs), we notably include "dark" hlePFCs (protein clusters without functional and taxonomic annotation below the Phylum rank), which are discarded and ignored in the majority of studies.

In addition, our study is the first to ask the question of the predictability of microbial composition from the environmental context, and to suggest insights. To highlight this aspect, we added Figure 1B to

the revised manuscript, which illustrates how models associated with PFCs highly linked to the environment offer the opportunity to make predictions at global scale using “only” 93 samples. In the revised version of the manuscript we put emphasis on these different points at lines 1027-1041.

2. predictability of protein functional clusters. Only ~1% of the detected protein functional clusters (2,444 out of 233,756) are predictable using the criterion in the study, questioning the applicability of machine learning and the subsequent major analyses based on these limited 2,444 protein functional clusters (Figure 2, 3 and 4). In this case, a substantial amount of data/information is ignored. For example, Mediterranean Sea was categorized as an outlier in these 2,444 protein functional clusters by the canonical correspondence analysis. But Mediterranean Sea may drive a large portion of the variation in PFCs (Figure 2, 3 and 4). Will it be worthwhile to re-do some of the analyses after removing data from the Mediterranean Sea and compare with current results? Besides, will it be worthwhile to conduct a canonical correspondence analysis on all of the 233,756 protein functional clusters or 44,653 protein functional clusters ($R^2 > 0.25$ in random forest model)?

Response: We deeply thank the reviewer for highlighting this point. We agree that the low proportion of PFCs included in the biogeographical analysis of the first version of the manuscript was limiting the strength and the extent of our results. In addition, the reviewer highlighted how negative R^2 values raised questions in his 5th detailed comment below. These two points pushed us to improve our approach of PFCs selection for this revised version.

First, we opted for a different formula of R^2 values calculation. In the precedent version of the manuscript, R^2 values were computed as:

$$R^2 = 1 - \frac{\text{sum}(\text{observations} - \text{predictions})^2}{\text{sum}(\text{observations} - \text{mean}(\text{observations}))^2}$$

Using this formula, model predictions that are further away from observations than when predicting the mean of the training set lead to R^2 values reaching values below 0.

In the new version of the manuscript, we switched for the Pearson-correlation based formula, more commonly used in biology;

$$R^2 = \text{correlation}(\text{observations}, \text{predictions})^2$$

Using this formula, R^2 values were bounded between 0 and 1, and the number of PFCs associated with models showing R^2 values over 0.5 increased from 2,444 (1.0% of total PFCs) to 14,585 (6.2% of total PFCs), allowing us to integrate more PFCs in our biogeographical analysis while retaining the same threshold of R^2 selection (please note that the addition of size fraction as a predictor, asked by reviewer #1, and the improvements made on our machine learning pipeline might also have impacted the number of models with R^2 values over 0.5). The updated results are extremely similar to the ones presented in the first manuscript in terms of global biogeographical patterns (e.g. Mediterranean and Polar samples remained outliers, distribution of samples in the canonical correspondence analysis (CCA) space was overall similar and very much reflecting Longhurst provinces, Figure 2). We believe that this demonstrates the robustness of our results and conclusions.

Second, we now provide a similar biogeographical analysis using a different threshold of R^2 selection, like proposed by the reviewer #2 (i.e., $R^2 > 0.25$). In section II of the supplementary results, we present a canonical correspondence analysis (CCA) exploring the biogeography of the 130,650 PFCs associated with models showing R^2 values above 0.25. This analysis includes 55.9% of the PFCs defined in our study, yet the general biogeographical patterns observed using a higher threshold of R^2 values (i.e. 0.5) were conserved (Figure S5). In addition to these conserved patterns, two samples from Station 93 were identified as isolated from the other samples along the first dimension of the CCA space, while they were not identified as outliers in the CCA based on hlePFCs only. In the revised manuscript, we now discuss this result, and show that this pattern can be explained by the fact that Station 93 is one of the rare coastal stations of the dataset (i.e. the dataset is mainly constituted of open-ocean stations), moreover located in an upwelling zone on the central-southern Chilean coast. Additional analysis (Figure S6, S7) confirmed the originality of these samples and

pointed out an overabundance of MAGs assigned to the *Pseudoalteromonas* genus. These new results are presented and discussed in the main text (e.g. lines 728-736 and 926-945) and the section II of supplementary materials in the revised manuscript.

Finally, we decided to keep Mediterranean samples in all the analysis presented in the manuscript. As highlighted by the reviewer #2, Mediterranean samples were clearly identified as outliers, potentially driving a large proportion of the explained variance in our CCA. However, the variance explained by their associated axis in the CCA space (CCA1<0 in the revised manuscript) was of 15.38%. Hence, about 85% of the variance in hlePFCs abundance data is either unlinked or poorly linked to the outlying position of Mediterranean samples. To confirm this, we computed a CCA on the 14,585 hlePFCs, this time excluding Mediterranean samples (detailed results not shown). It led to a decrease of 0.8% in adjusted R^2 compared to the CCA performed on all samples (67.4% against 68.2% when using all samples). The first two axes of the CCA excluding mediterranean samples explained 27.93% of the variance, against 28.55% in the complete CCA.

In the CCA performed on PFCs associated with models satisfying the $R^2 > 0.25$ threshold, the third axis was associated with the outlying position of Mediterranean samples, and its percentage of explained variance was 6.80% (section II of the supplementary results). In comparison, the first axis, associated with the newly observed pattern showing Station 93 as an outlier, had an explained variance of 12.75%. This demonstrates how the identification of Mediterranean samples as outliers, although clear and conserved across all analyses, was not responsible for the overall significance of our statistical analysis.

Our decision of keeping Mediterranean samples in the revised version of the study was finally motivated by the fact that removing these samples from all statistical analysis would reduce the number of samples from 93 to 86. This leads to a reduction of 9% in the size of training sets. We believe that such a reduction would make it difficult to conduct an unbiased comparison of the results from our machine learning approach with and without Mediterranean samples (*i.e.* distinguishing the effects of removing Mediterranean samples from the ones of reducing training sets sizes).

We hope that these arguments will convince the reviewer and editor, especially considering the fact that an increased number of PFCs has been included in the revised version of the study, allowing to put the impact of Mediterranean samples in a broader perspective.

3. clarity of the method and results description. This is really a comprehensive analysis on the proteins retrieved from 885 MAGs. Starting from 1,914,171 proteins to 233,756 connected components and then to 2,444 "hlePFCs", there are so many numbers and percentages in Results section in addition to Tables 1 and 2. It may be useful to have a flowchart to illustrate how to produce the different group of data (size of the data) and the exact analyses done in each step to these data. For example, 233,756 connected components were selected based on sequence similarity network and then 228,914 clusters were chosen due to the non-zero variance abundance profiles.

Response: We followed the advice from the reviewer, and now provide a flowchart of our methods in Figure S1, presenting the different datasets, their sizes and how they were obtained.

Here are the detailed comments:

1. Line 1: "predictions" used in the title may not be appropriate. This study mainly focuses on the correlation analysis between planktonic functions and environmental data. It does not provide any further predictions at the regional or global scale.

Response: The amount of samples in our dataset did not allow us to extrapolate global scale predictions, and a large part of our focus was thus aimed towards a biogeographical analysis of PFCs' distribution. Extrapolating our results to build predictions in places that had not been sampled during *Tara Oceans* cruises would have been unjustified at this point, considering the novelty of our

approach. However, we do make predictions of PFCs abundance in this study, both based on out-of-bag samples and test sets, and we prove that such predictions are accurate for at least 6% of our data (setting a threshold of $R^2 > 0.5$). To insist on the fact that abundance predictions do constitute a primordial step in our pipeline, we added Figure 1B to the revised manuscript, showing the correlation between abundances observed in test sets and their associated model predictions. To our knowledge, our study is the first to test the predictability of PFCs' abundance in the global ocean. Testing this predictability on test sets appeared as the obligate first step towards a future possibility of extrapolating abundance predictions. Our methodology will first have to be tested on different samples, from different sampling events, focusing on different ecosystems, before computing extrapolations. We took thus the precaution of entitling our paper "Towards omics-based predictions..." and not "Omics-based predictions...", although omics-based predictions were performed in our pipeline.

2. Line 129: *Protein functional cluster (PFCs) instead of in line 131*

Response: Thank you, corrected.

3. Line 131: *"numeroted" to "numbered" or "enumerated"*

Response: Thank you, corrected.

4. Line 184: *Since Longhurst biogeographical provinces were identified as the best predictor, it may be useful to show the Longhurst biogeography map with sampling locations on top of it in the supplementary. It would help to visualize the Figure 2 legend that only shows the acronyms of these provinces.*

Response: We would like to thank the reviewer for his advice. As we already provide 7 supplementary figures in the revised version of the manuscript, we decided to add the map of Longhurst provinces directly in Figure 2, and not in the supplementaries. We agree that it helps to better visualize the Figure 2 legend.

5. Figure 1. *It would be nice to show at least one example of the random forest model (observed versus predicted abundances in the training and test datasets) and explain the negative R2 from a large number of models. Try to keep the notations of environmental variable consistent in Figure 1B and in the Table S2. For example, temperature is described in the text while temperature anomaly is shown in the x axis of Figure 1B.*

Response: As stated in our response to the reviewer's 2nd major comment, this detailed comment was of great use to us, and pushed us to improve our approach concerning PFC selection. Figure 1B now gives an illustration of random forest predictions on test sets for all models with R^2 values above 0.5. Concerning the reviewer's last comment, temperature (measured on site) and temperature anomaly (obtained from climatology data) are both represented in Figure 1C (previously Figure 1B).

6. Lines 199 to 204, *could you explain why only a small fraction of the exact same metabolic pathways are predictable?*

Response: Our hypothesis is that only small fractions of the same metabolic pathways are predictable because most metabolic pathways are quite ubiquitously present in the ocean, but sustained by different organisms and protein families. Our results show that protein functional clusters highly linked to the environment (hlePFCs) belong to groups of organisms associated with particular environmental contexts (e.g. cold polar waters, saline Mediterranean samples,...), without being related to functions specific to these contexts. Hence, for a single metabolic pathway, PFCs can exhibit very different

levels of predictability, depending on the level of association between their organisms of origin and particular environmental conditions. In addition, it appeared to us that the ranking of “most selected” pathways, previously presented at lines 199 to 204, was quite sensitive to the thresholds of (1) minimum number of occurrences in the network to be considered in the ranking (arbitrarily fixed at 1000 in the original manuscript), and (2) minimum R^2 value to be considered as an hlePFC. Thus, in the revised manuscript the content of lines 199 to 204 was removed, while Table S1 gives the number of occurrences and the percentage of selection among hlePFCs for all pathways.

7. Line 208: “was divided by two” to “was halved”

Response: Thank you, corrected.

8. Lines 242-250: *Instead of categorizing Mediterranean Sea as the exception, isn't Mediterranean Sea explaining a large fraction of the variation of hlePFCs (see a large number of hlePFCs are located in the CCA2>2)? It would be interesting to do all the analyses and see what the results look like after removing data from the Mediterranean Sea. How many hlePFCs will exist and how will the CCA plot look like?*

Response: As explained above (2nd major comment of the reviewer #2), we chose to keep Mediterranean samples in all analysis. CCA1, the axis associated with Mediterranean samples, explained 15.38% of the total variance in our data. Also, excluding the Mediterranean samples from our machine learning approach would modify the number of hlePFCs not only because many PFCs were linked with Mediterranean samples, but also because the size of training sets would be reduced by 9%. It would thus be difficult to draw conclusions on the impact of excluding Mediterranean samples through this approach.

9. *Figure 2, it would be interesting to color-code the hlePFCs with R^2 values and see which dimension presents hlePFCs with high R^2 values.*

Response: We would like to thank the reviewer for this interesting idea. We tried to implement it in the new version of the manuscript, however adding a layer of color underneath Figure 2 makes it really hard to read. We present below an additional figure only representing hlePFCs color coded according to their R-squared values in the CCA space, which could be added in the supplementary files. But no surprising pattern appeared on this Figure, as R-squared only tended to be higher in the periphery of the CCA space, notably in Mediterranean and Polar samples, already identified as the strongest outliers. Thus we decided not to include the Figure in the revised manuscript or the supplementary files.

Additional Figure : Distribution of the 14,585 hlePFCs in the CCA space, as pictured in Figure 2 of the main text. Each point corresponds to a PFC, and is colored according to the R^2 value of its associated random forest model.

10. Line 300, Atlantic North-West is not shown in Figure S3.

Response: Thank you, corrected.

11. Lines 338-339, O2 doesn't show a higher rank than temperature in Figure 2B.

Response: The Figure has changed, oxygen is now at a higher rank than both temperature anomaly and temperature, while before it was below temperature anomaly and above temperature.

12. Lines 366-368 and line 19.5% and 1.1%, it would be nice to note how these numbers are derived, e.g. (2,444/233,756).

Response: These numbers have been updated and we now indicate how they are derived in lines/section 824-825.

13. Lines 387-395, they are still descriptive not explanatory. For example, why methane metabolism is one of the most selected pathways in hlePFCs while not the nitrogen metabolism? It would be good to provide some discussions.

Response: As exposed in our response to point #6, we avoided drawing conclusions from the ranking of most selected pathways in the revised version of the manuscript, focusing more on the links between pathways and environmental conditions than on their overall selection among hlePFCs.

14. Lines 396-397, again it would be interesting to run some analysis after removing the data from Mediterranean Sea. This helps to double-check whether Mediterranean Sea samples are only minor outliers or are actually driving the global pattern.

Response: As stated before, the outlying position of Mediterranean samples is responsible for a maximum of 15.38% of the variance in PFC abundances, but probably less considering that the CCA1 axis also allows to distinguish subtropical samples of the Indian Ocean. We then consider Mediterranean samples as outliers on a level quite comparable to the ones from the Southern Ocean in our CCA space, while not being solely responsible for the global pattern.

15. Line 416 and the following section, it is an originality that this study considers both annotated and unannotated sequences. But it would be good to provide some additional valuable information from this discussion in addition to suggestions for future work. Only 0.3% of the microbial dark matter (24 out of 7834) is selected as hlePFCs. What about the other unpredictable microbial dark matter?

Response: Based on the reviewer's comment, we added elements in the discussion section concerning the unpredictable dark PFCs: "In addition, more than half (52.9%) of the models associated with dark PFCs showed R^2 values above 0.25, and the mean R^2 values over the 7,834 models associated with dark PFCs was similar to the one over all models. We then show that the response of dark PFCs to environmental gradients is comparable to the one of taxonomically and functionally annotated PFCs."

Methods

1. Line 486, as mentioned above, it might be clearer to read the Figure 2 if showing the sampling/data locations on top of the Longhurst provinces in a supplementary figure.

Response: Figure 2 now includes a map of Longhurst provinces.

2. Lines 606-612, how is 75% of the sample selected for each training datasets? Did you have a chance to look at the results of the out of bag samples or validate the model in the test dataset to evaluate the model performance?

Response: Training sets were defined through random draws, using the createDataPartition function from the Caret R package. We thank the reviewer for his comment on out-of-bag samples, which were used to compute R^2 values although it was not clearly exposed in the first submitted version of this manuscript. We now provide in Figure 1A two measures of R^2 values, one based on the out-of-bag samples from our cross validation, and one based on predictions on our test sets. We also provide a more detailed explanation of our approach in the Methods section.

3. Line 643-644, it would be good to provide a text description for each file in this link <https://figshare.com/s/b33fc72a62db44b7192f>.

Response: We added the sentence "including fasta files containing nucleotide sequences of all proteins in PFCs, hlePFCs, dark PFCs, PFCs associated with the Station 93, and PFCs associated with the Station 93 linked with Pseudoalteromonas MAGs. In this figshare repository, we also provide summary tables including all PFC and random forest associated statistics (e.g. all homogeneity and unknown scores, R^2 values, variables importances) for each PFCs, hlePFCs and dark PFCs. Finally, we offer tables at the single protein level showing the PFC ID, taxonomic and functional annotations, and nucleotide sequences of each protein in PFCs, hlePFCs and dark PFCs." (lines 1236-1301). We decided not to provide a detailed file by file description here as such descriptions are already available on the figshare page itself, which contains a total of 21 files. We thought that describing each 21 files

would make the *Data availability* section quite long. If however the reviewer and editor believe it is necessary to include a file by file description in this section, we will add one.

References:

Righetti, D., Vogt, M., Gruber, N., Psomas, A., & Zimmermann, N. E. (2019). Global pattern of phytoplankton diversity driven by temperature and environmental variability. *Science advances*, 5(5), eaau6253.

Salazar, G., Paoli, L., Alberti, A., Huerta-Cepas, J., Ruscheweyh, H. J., Cuenca, M., ... & Gregory, A. C. (2019). Gene expression changes and community turnover differentially shape the global ocean metatranscriptome. *Cell*, 179(5), 1068-1083.

Sunagawa, S., Coelho, L. P., Chaffron, S., Kultima, J. R., Labadie, K., Salazar, G., ... & Cornejo-Castillo, F. M. (2015). Structure and function of the global ocean microbiome. *Science*, 348(6237), 1261359.

Reviewer comments, second round –

Reviewer #1 (Remarks to the Author):

I would like to thank the authors on how they have improved the manuscript overall, according in part to my comments, and especially the small concern regarding size fraction differences between oceanic regions. It is now clear that this has a very limited effect on the analysis.

I have no other comments regarding this manuscript, which I found very interesting, and mature.

Best

Tom O. Delmont

Reviewer #2 (Remarks to the Author):

In the revised manuscript, the authors have adequately addressed the previous comments, e.g., the clarity of the method, how different inclusions of protein functional clusters affect the analysis/prediction and the anomaly of Mediterranean samples. Here are a few additional comments.

Line 363-365. When including PFCs with $R^2 > 0.25$, Figure S5 (Figures S5-S7) actually does not show the similar patterns as Figure 2 (Figures 2-4). This reflects the sensitivity of such analysis to the number of samples or PFCs included. It may be more appropriate to mention the differences as discussed in the supplementary material: "It highlights how the potential of our approach to pinpoint PFCs associated with particular environmental conditions is highly dependent on the amount of samples included, and the diversity of conditions that they represent."

Figure 1B. Because there are so many points in the scatter plot, it may be clearer to color-code the density of the points, which shows positions where most data points are distributed.

Figure S5. It may be useful to label the two outlier samples/points from station 93.

Reviewer #1 (Remarks to the Author):

I would like to thank the authors on how they have improved the manuscript overall, according in part to my comments, and especially the small concern regarding size fraction differences between oceanic regions. It is now clear that this has a very limited effect on the analysis.

I have no other comments regarding this manuscript, which I found very interesting, and mature.

Best

Tom O. Delmont

Reviewer #2 (Remarks to the Author):

In the revised manuscript, the authors have adequately addressed the previous comments, e.g., the clarity of the method, how different inclusions of protein functional clusters affect the analysis/prediction and the anomaly of Mediterranean samples. Here are a few additional comments.

Line 363-365. When including PFCs with $R^2 > 0.25$, Figure S5 (Figures S5-S7) actually does not show the similar patterns as Figure 2 (Figures 2-4). This reflects the sensitivity of such analysis to the number of samples or PFCs included. It may be more appropriate to mention the differences as discussed in the supplementary material: "It highlights how the potential of our approach to pinpoint PFCs associated with particular environmental conditions is highly dependent on the amount of samples included, and the diversity of conditions that they represent."

We understand the concern of the reviewer, and agree that the match between patterns observed on Figure 2 and S5 is not perfect. In fact, the patterns observed on Figure 2 are well duplicated on the third and fourth axis of the CCA based on the 130,651 PFCs linked to models showing R^2 values above 0.25 (Figure S5B), but not on the first and second axis (Figure S5A). The text was modified to be clearer on this point:

"A CCA based on these 130,651 PFCs allowed to identify 2 samples from the surface and deep chlorophyll maximum of station 93 as strong outliers due to their singular position along CCA1 (Supplementary Figure 5A). The outlying nature of these two samples was explained by the strong overabundance of PFCs composed of proteins from 5 MAGs of the genus *Pseudoalteromonas* (Supplementary Figure 6, Supplementary Figure 7). The general patterns observed using the 14,585 hlePFCs (Figure 2) were confirmed by the third and fourth dimensions of the CCA on 130,651 PFCs (Supplementary Figure 5B)."

Figure 1B. Because there are so many points in the scatter plot, it may be clearer to color-code the density of the points, which shows positions where most data points are distributed.

We thank the reviewer for proposing this improvement on Figure 1B. We decided to represent hexagonal bins colored according to the number of points found in each of them, as it led to a clearer Figure than a scatter plot colored according to density.

Figure S5. It may be useful to label the two outlier samples/points from station 93.

We thank the reviewer for pointing this out, the two samples were highlighted by black arrows on Figure S5.